# Geometry of naturalistic object representations in recurrent neural network models of working memory

Xiaoxuan Lei[1,2]               Takuya Ito[3]

Pouya Bashivan[1,2]

[1] Department of Physiology, McGill University, Montreal, Canada
[2] Mila, Université de Montréal, Montreal, Canada
[3] IBM Research, Yorktown Heights, NY, USA

## Abstract

Working memory is a central cognitive ability crucial for intelligent decision-making. Recent experimental and computational work studying working memory has primarily used categorical (i.e., one-hot) inputs, rather than ecologically-relevant, multidimensional naturalistic ones. Moreover, studies have primarily investigated working memory during single or few number of cognitive tasks. As a result, an understanding of how naturalistic object information is maintained in working memory in neural networks is still lacking. To bridge this gap, we developed sensory-cognitive models, comprising of a convolutional neural network (CNN) coupled with a recurrent neural network (RNN), and trained them on nine distinct N-back tasks using naturalistic stimuli. By examining the RNN's latent space, we found that: 1) Multi-task RNNs represent both task-relevant and irrelevant information simultaneously while performing tasks; 2) While the latent subspaces used to maintain specific object properties in vanilla RNNs are largely shared across tasks, they are highly task-specific in gated RNNs such as GRU and LSTM; 3) Surprisingly, RNNs embed objects in new representational spaces in which individual object features are less orthogonalized relative to the perceptual space; 4) Interestingly, the transformation of WM encodings (i.e., embedding of visual inputs in the RNN latent space) into memory was shared across stimuli, yet the transformations governing the retention of a memory in the face of incoming distractor stimuli were distinct across time. Our findings indicate that goal-driven RNNs employ chronological memory subspaces to track information over short time spans, enabling testable predictions with neural data.

## 1 Introduction

Working memory (WM) – the ability to store and manipulate information over short periods – is a central cognitive capability that enables a wide spectrum of behaviors [Baddeley, 1992]. Over the past few decades, various experimental, computational, and theoretical techniques have been adopted to study WM from both cognitive and neural perspectives. However, several fundamental issues remain unresolved, including the key question of *how high-dimensional sensory information is encoded, maintained, and modulated according to specific task demands*.

A significant body of work has been dedicated to modelling the computations underlying WM. These include many classic models from cognitive science [Meyer and Kieras, 1997a, Ritter et al., 2019,

38th Conference on Neural Information Processing Systems (NeurIPS 2024).

Baddeley et al., 2021], neuroscience [Miller et al., 1996, Wang, 1999, Emrich et al., 2013], and more recently deep learning [Yang et al., 2019, Ehrlich and Murray, 2022]. Recent approaches have focused on using artificial neural networks, such as recurrent neural networks (RNNs), to understand and model WM due to their ability to learn the complex cognitive tasks that are commonly used to study WM in humans. However, prior studies were limited in three aspects. First, most works used abstract categorical inputs, such as color and location of moving dots that are represented as binary (or one-hot vector) inputs [Panichello and Buschman, 2021, Yang et al., 2019, Piwek et al., 2023]. While training models with such categorical inputs is easier in practice, the resulting models offer limited insights into how real world, naturalistic stimuli (which are embedded in high-dimensional spaces) are processed. Second, most prior work has considered single or a few cognitive tasks [Xie et al., 2022, Mante et al., 2013, Piwek et al., 2023], limiting the generality of these models in explaining the neural computations underlying working memory. Lastly, while some prior work has explored how object features are encoded into population activity [Xie et al., 2022], it remains unclear how information in working memory is sustained across time to support concurrent encoding, retention, and retrieval during a dynamic task—and how these processes might align with classic cognitive theories [Luck and Vogel, 1997, Alvarez and Cavanagh, 2004, Franconeri et al., 2013].

To address these limitations, we investigated how task-optimized RNNs manipulate the multidimensional properties of naturalistic visual inputs during different stages of WM (Figure 1d). Specifically, we examined: 1) How do task-optimized RNNs select task-relevant properties of naturalistic objects during WM? 2) What computational strategies do RNNs employ to dynamically maintain object properties in the face of incoming (distractor) information? To address these questions, we trained *multi-task models on a collection of N-back tasks using naturalistic stimuli*, and analyzed the RNNs' latent space during concurrent encoding, retention, and retrieval of information. We specifically chose the N-back task, given that it requires the dynamic encoding, retention, and retrieval of information in the face of incoming distractor information. This is in contrast to prior studies, which primarily studied WM tasks such as delayed-match-to-sample tasks, which focus on WM maintenance during stable fixation periods. The nature of the N-back task makes it an excellent testbed to evaluate how naturalistic object features are maintained in a dynamic environment that requires on-the-fly WM updates and decision-making.

Our main contributions are as follows:

- We trained different classes of gated and gateless RNN models on a suite of WM tasks and developed decoder-based analyses to study the geometry of naturalistic object representations during different stages of WM.

- We found that task-relevant and -irrelevant object properties are simultaneously encoded in multi-task RNN models, while only gateless RNNs produced shared and reusable representations across tasks.

- We found that object features are less orthogonalized in the RNNs' hidden dynamics compared to perceptual representations.

- We found that RNNs solve the N-back task by using chronological memory subspaces to separate object representations presented across time. This finding supports resource-based models of working memory [Alvarez and Cavanagh, 2004, Franconeri et al., 2013] and challenges the classic slot-based model [Luck and Vogel, 1997].

## 2    Related Works

**Models of working memory.**    Originally rooted in cognitive science, the notion of WM was first formally defined and popularized by Baddeley [1992] who proposed a cognitive system consisting of modality-specific buffers and a shared executive module to control the information flow in and out of the memory buffers. This initial work, along with most early models, portrayed WM as a memory system with three key features: flexibility of information representation, limited capacity, and limited temporal span.

Subsequent models based on this perspective were largely akin to the architecture of the Von-Neumann computers, comprising of input and output channels, volatile memory components, and a central processing system that continuously executed pre-specified computer code according to task goals [Cowan, 1988, Meyer and Kieras, 1997b, Anderson, 2013]. However, experimental work in

neuroscience has showed that the underlying biological circuitry of WM consists of a wide network of brain areas with diverse roles that do not adhere to the clean-cut modules specified in those previous models [Sreenivasan and D'Esposito, 2019].

Compte et al. [2000] was one of the first studies that showed RNNs with closely matched connection parameters to those measured from the brain can reliably store information in the presence of distractors. Later work showed that these networks are not only capable of performing many classic WM tasks but also replicate neural signatures that were previously observed in animals' brains during these tasks, indicating that similar neural computations may be used by both systems [O'Reilly and Frank, 2006, Mante et al., 2013, Xie et al., 2023, Finkelstein et al., 2021, Masse et al., 2019]. More recent work proposed various neural network architectures that combine linear attention mechanisms with slot-based memory modules and feedback mechanisms as models of working memory [Hwang et al., 2024, Loynd et al., 2020].

**Neural network models in neuroscience.** Neural network models have been increasingly used in computational neuroscience to model neural computations during different behaviors. These models are valuable tools that incorporate ideas from a diverse range of cognitive architectures, while also making testable behavioral or neural predictions. Various classes of neural network models have now been used to simulate neural activity in sensory brain regions [Schrimpf et al., 2018, Kell et al., 2018, Khaligh-Razavi and Kriegeskorte, 2014, Yamins et al., 2014, Bashivan et al., 2019], language comprehension [Hasson et al., 2020, Schrimpf et al., 2021], and decision making [Mante et al., 2013, Yang et al., 2019]. Furthermore, others have used RNNs to study the dynamical motifs that underlie various behavioral signatures such as memorization, integration and selection of information, and attention. For example, Mante et al. [2013] used goal-directed RNNs to simulate the dynamics of neural populations in the macaque prefrontal cortex during a context-dependent decision making task and found specific dynamical mechanisms for the selection and integration of task-relevant inputs. Yang et al. [2019] identified specialized functional clusters, mixed selectivity, and compositional representation in multi-task RNNs. The shared dynamic motifs across tasks were further extended and described mathematically in Driscoll et al. [2022].

## 3    Methods

**Tasks.** We considered N-back tasks ($N \in \{1, 2, 3\}$) based on one of three distinct object properties (i.e. feature; $F \in \{Location, Identity, Category\}$ (denoted as $L, I, C$), resulting in a total of 9 N-back task variants (Figure 1b). Naturalistic stimuli were generated using 3D object models from the ShapeNet dataset (rendered examples in Figure A1a) [Chang et al., 2015], comprising 4 object categories, each with 2 unique identities rendered from various view angles, and presented at 1 of 4 possible locations. We consider two validation approaches: validating on novel view angles, and validating on novel identities. The training and validation novel angle datasets differed in their viewing angles, necessitating view-invariant processing by the model. In contrast, the validation novel identity includes unseen identity sampled from categories same as the training dataset.

**Model Architecture.** We considered a two-stage model that delineates perceptual and cognitive processes (Figure 1c). At the first stage, the model processes sequences of images, utilizing an ImageNet [Deng et al., 2009] pre-trained ResNet50 [He et al., 2016] model to derive visual embeddings from each image input. All object features including category, identity, and location were highly decodable from these activations (category: 100.00%, identity: 99.57%, location: 100.00%; 2-fold cross-validation). A point-wise convolutional layer reduces the dimensionality of the 2048-channel feature map from ResNet's penultimate layer (layer 4.2 ReLU) to match the RNN's hidden size. Next, the vectorized embeddings are concatenated with a task index vector and processed by a fully connected layer (matching the RNN's latent size) with layer normalization [Ba, 2016], serving as input to the RNN. The RNN's output is then fed through another fully connected layer and projected to one of three possible responses: match, non-match, or no action.

Each network is trained to perform one (single-task-single-feature) or multiple tasks (multi-feature or multi-task or both). After training, we analyzed activations from the penultimate layer of ResNet50 (i.e. *the perceptual space*), as well as the RNN activations during the stimulus presentation and subsequent timesteps (i.e. *encoding and memory space* respectively, denoted as $E$ and $M$ as shown

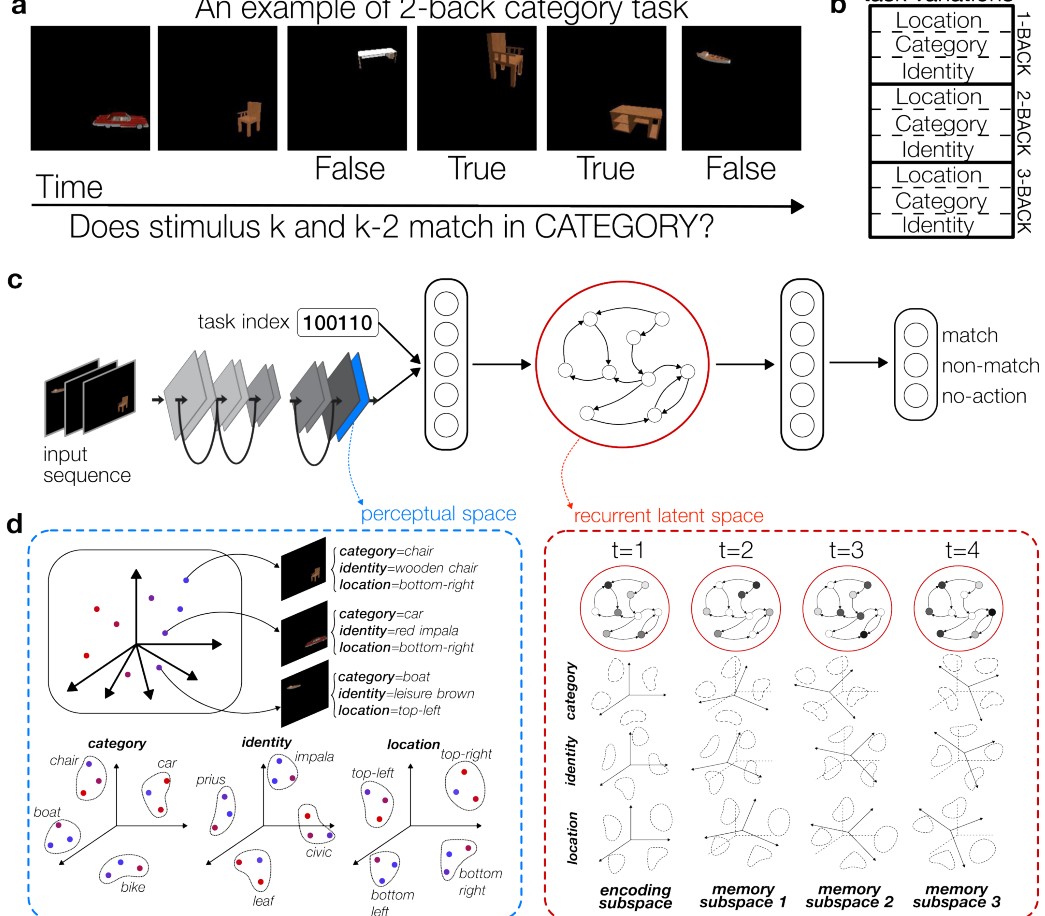

Figure 1: **Tasks and Models: a)** Example of a 2-back category task. Each object's category is compared with the category of the object seen two frames prior. **b)** The suite of n-back tasks considered in the study. **c)** The sensory-cognitive model architecture. **d**) A schematic showing the latent subspaces for category, identity, and locations in the perceptual, encoding, and memory subspaces. Left: Stimuli are encoded in high dimensional latent space of the vision model (CNN). Each object property is encoded in a high dimensional latent subspace of this model; Right: RNN model represents each object property in its encoding latent subspace and retains some or all of the properties within its memory subspaces at later time points.

in Figure 4a). We considered three recurrent architectures for the second stage of the model including the vanilla RNN, GRU [Chung et al., 2014], and LSTM [Hochreiter and Schmidhuber, 1997].

**Model Training**   Each model architecture was trained within three scenarios that differed in their training diet:

- **Single-Task-Single-Feature (STSF)**: trained on a single n-back task based on a single object feature (e.g.1-back location).
- **Single-Task-Multi-Feature (STMF)**: trained on a single choice of $N$ for all three feature variations (e.g. 1-back $L$, or $C$ or $I$).
- **Multi-Task-Multi-Feature (MTMF)**: encompassing all choices of $N$ (1,2,3) and features $(L, I, C)$.

Model parameters were trained using AdamW [Loshchilov and Hutter, 2017] optimizer with an initial learning rate of $3e-5$ and a Multi-step learning rate decay with $\gamma = 0.1$ for every 100 iterations. Batch size was 256 and all trials were generated on-the-fly using the iWISDM package [Lei et al.,

2024]. Hidden layers of RNN modules are initialized with Kaiming Initialization [He et al., 2015]. For details of decoding analysis and Procrustes analysis, please refer to Appendix .1

# 4 Experiments

All models reached $> 95\%$ accuracy on train and $> 90\%$ on validation set with novel object angles. Generalization to novel object instances was substantially weaker (Figure A1b, c). Model performance increased with the number of model parameters, and with identical parameter count, vanilla RNNs accuracy was lower compared to their gated counterparts (Figure A1c). The ensuing analyses utilized data collected from models with 512 units for vanilla RNNs, and 256 units for GRUs and LSTMs, to ensure comparable model performance as well as comparable model parameters.

## 4.1 Encoding of task-relevant and -irrelevant object properties in task-optimized RNNs

We first examined how RNN modules represent various object properties such as location, identity and category in their latent space. In particular, we investigated the following two questions:

**1) Do recurrent networks preserve object properties that are not necessary for the task?** In order to perform a task, recurrent networks must maintain information about the task-relevant object properties. However, maintaining information about the task-irrelevant factors is not necessary from the perspective of the task objective. Therefore, RNNs may either selectively maintain task-relevant information or maintain full object representations, recalling task-relevant information when prompted.

We trained decoders (i.e. classifiers) to predict each object property from the RNN hidden state activity from the first timestep of each trial (e.g. $F = L_i$ vs. $F = L_{j \neq i}$ decoders, total 4 location decoders). Cross-validated decoding accuracies are shown in Figure 2b for STSF, STMF, and MTMF GRU models. Unsurprisingly, the task-relevant object properties are fully retrievable in all models. Further tests revealed a causal relationship between the subspaces encoding task-relevant information and the network's generated response (Fig. A7; see section .1 for details.) However, while task-irrelevant object features are not generally well-preserved in STSF models (Figure 2b, left), they are much better preserved in STMF and MTMF models (i.e. decoding accuracy $> 85\%$; Figure 2b middle and right). This finding was consistent across all three RNN architectures (Figure A2a) and across time points (Fig. A5). These results suggested that all RNNs maintained a full representation of objects in their latent spaces regardless of which object properties were required for performing the task.

**2) Are object properties encoded within a subspace that is shared across different tasks or distinct ones within each task?** Having observed that both task-relevant and -irrelevant information are retained by multi-feature RNNs, we next asked whether RNN encoding of object properties is task-dependent or -independent. To probe this, we trained decoders to predict object properties from the RNNs' activations during one task, and tested the decoder on RNN activations when performing another task (i.e., cross-task decoding). We quantified the generalization performance of MTMF models across all three architectures (Figure 2a). We found that gated RNNs (GRU and LSTM, Figure A2b) utilized highly task-specific subspaces for encoding object properties, while vanilla RNN encoded object properties within a subspace that was shared across all task-variations (Figure 2c). This suggests that gated RNNs tend to learn task-specific representations that do not generalize across tasks, potentially impacting their ability to generalize to new tasks.

## 4.2 Representational orthogonalization in task-optimized RNNs.

To improve their performance, RNN weights might form structured and separable representations for each task-relevant feature. For this to be true, the RNN latent space may orthogonalize feature representations beyond their perceptual representation (see schematics in Figure 3a). To quantify orthogonalization, we calculated the angles between all pairs of decision hyperplanes using cosine similarity (bootstrapped 10 times). We then summarized these angles into a single orthogonality measure by computing the Frobenius norm of the difference between this matrix and the identity matrix (which represents complete orthogonalization). We defined this measure as the orthogonalization

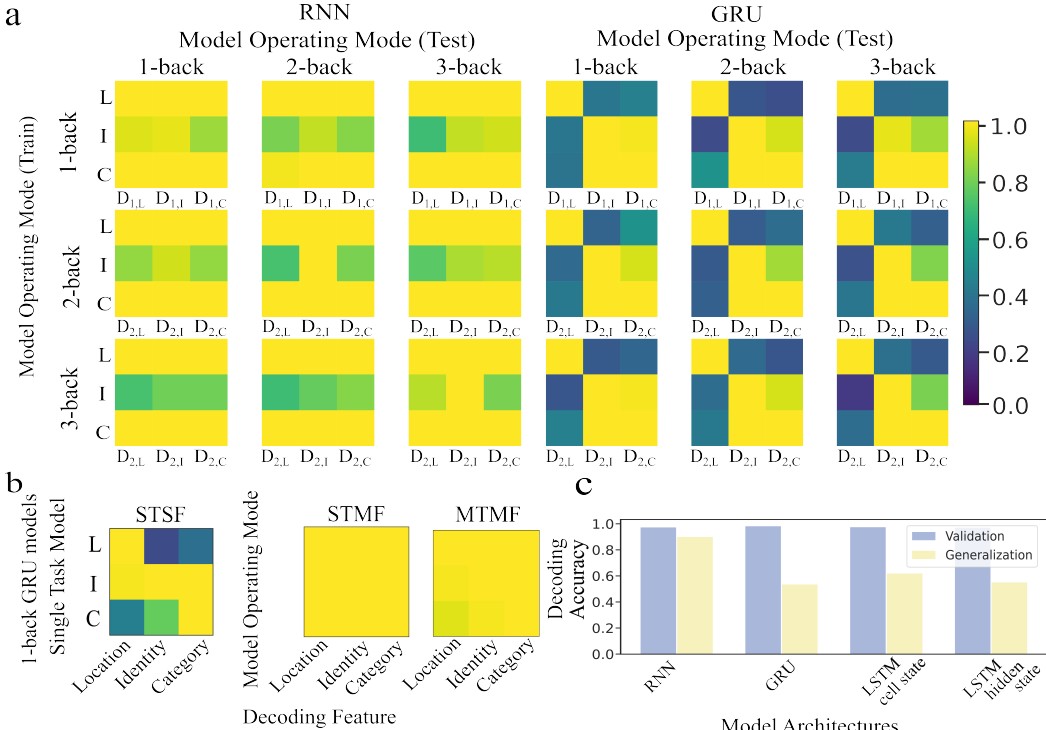

Figure 2: **Representation of task-relevant/-irrelevant object properties:** **(a)** Decoding generalization accuracy for each object property is displayed across tasks and operating modes for vanilla RNN and GRU. Rows and columns of $3 \times 3$ matrices correspond to the $N$-back task on which the decoders are fitted and tested on respectively. Matrix columns correspond to particular decoders denoted by $D_{k,F}$ ($k \in \{1, 2, 3\}$, $F \in \{L, I, C\}$) (indicating which task and decoding feature the decoder was fitted on), while matrix rows correspond to the object property of the task the decoder was tested on. **(b)** Validation accuracy of decoders trained on RNN latent space activations from the first time step of each trial to predict different object properties. Each column represents the object property the decoder was trained on, while each row corresponds to a model. **c)** Quantification of the validation accuracy (within the same task, indicated in purple) and generalization accuracy (across tasks with different task-relevant features, indicated in yellow) across all model architectures.

index ($O$):

$$\tilde{W}_{ij} = 1 - \text{abs}\Big(\cos(W_i, W_j)\Big)$$

$$O = \mathbb{E}\Big(\text{triu}(\tilde{W})\Big) \tag{1}$$

where $W_i$ is the normal vector of the decision hyperplane that separates points assigned with feature value $i$ from the rest, $\cos(W_i, W_j)$ is the cosine similarity between the two normal vectors. We take the absolute value since the relative direction does not matter. $\text{triu}(.)$ is the upper triangle operator.

We evaluated the degree of representation orthogonalization within the perceptual and encoding spaces. Contrary to our hypothesis, we observed that relative to the perceptual space, the RNN latent space slightly de-orthogonalize the axes along which distinct object features are represented (Figure 3b). Similar results were found when PCA was used to equalize dimensionality between the two spaces (Figure A3). Although more orthogonalized representations generally facilitate structured and enhanced separation of task-relevant features, our behavioral results still indicate optimal performance. One possible interpretation of these results is that the reduced orthogonalization in the RNN latent space produces a more efficient (lower dimensional) representation. (In contrast, increased orthgonalization in the representational space would increase the overall dimensionality of the object representations.) In practice, only a subset of dimensions need to contain orthogonalized representations for successful task performance. In turn, this would make it easier to train the read out weights of the RNN to produce the correct task responses.

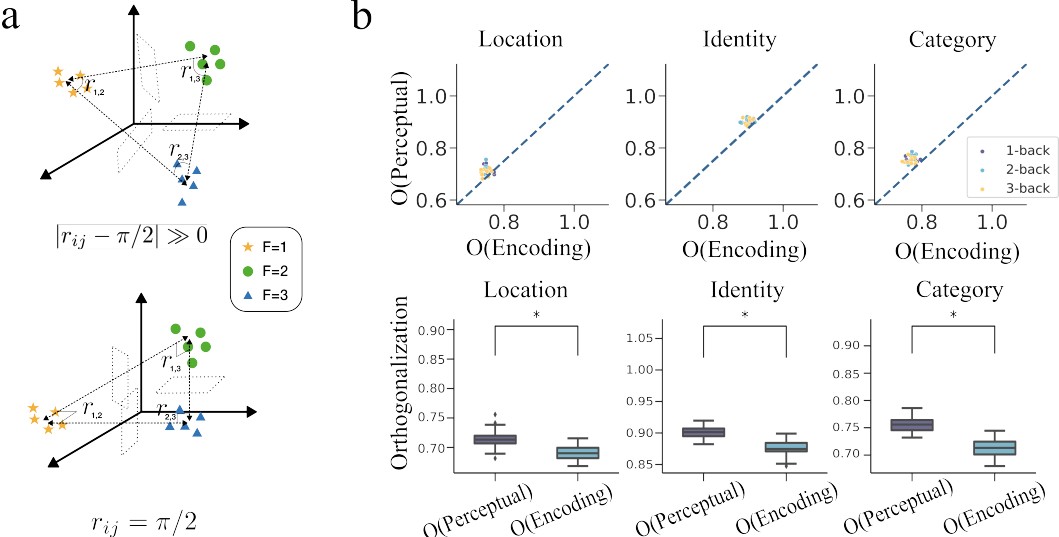

Figure 3: **Orthogonalization:** a) A schematic of two hypothetical object spaces in 3D. $r_{i,j}$ represents the angle formed by the decision hyperplanes that separate feature value $i$ and $j$ from each other. Top: non-orthogonalized representation; Bottom: orthogonalized representation. b) Upper panel: Normalized orthogonalization index, for both perceptual and encoding spaces respectively (denoted as $O(Perceptual)$ and $O(Encoding)$). In most models, a less orthogonalized representation of feature values emerges in the RNN encoding space compared to the perceptual space (CNN output). Lower panel: Statistical comparison of the relative orthogonalization levels between the perceptual and encoding spaces. A two-sample t-test was performed to assess differences between the distributions of orthogonalization indices in the perceptual space and the encoding space.

## 4.3 Neural mechanisms of concurrent encoding, maintenance, and retrieval in RNN models of WM

Having examined the encoding of objects in the RNN's latent space, we next investigated how RNN dynamics enable simultaneous encoding, maintenance, and retrieval of information. Performing our N-back task suite required the RNN to keep track of prior objects' properties while simultaneously encoding incoming stimuli with minimal interference.

We reasoned that the RNN may implement one of three possible mechanisms to perform the N-back working memory task suite (Figure 4e):

- **H1: Slot-based memory subspaces [Luck and Vogel, 1997].** Where the RNN latent space is divided into separate subspaces that are indexed in time within the sequence. Each object is encoded into its corresponding subspace (i.e. slot) and is maintained there until retrieved. By definition, the subspace assigned to each memory slot is distinct and "sustained" in time.

- **H2: Relative chronological memory subspaces.** Where the RNN latent space is divided into separate subspaces that each maintains object information according to their age (i.e. how long ago they were encoded). Such a mechanism will require a dynamic process for updating the content of each memory space at each time step during the task.

- **H3: Stimulus-specific relative chronological memory subspaces.** This is similar to the relative chronological memory hypothesis but with independent subspaces assigned to each object. Each observation in the sequence is thus encoded into a distinct subspace and encoding of each stimulus is in turn distinctly transformed into associated memory representations.

To identify the hypothesis that best matches the computations performed by the RNN, we analyzed how the RNN latent subspace encodes and transforms each object property ($E_{(S,T)}$) across time into memory ($M_{(S,T)}$) (Fig. 1d).

We first tested whether object information is maintained in a temporally stable subspace (i.e. sustained working memory representation) which aligns with **H1** prediction (i.e. $E_{(S=i,T=1)} \stackrel{?}{=} M_{(S=i,T=k)}$, $k \in \{2,3,4...\}$; Fig. 4a). For this, we trained decoders to predict the value of each object property using the RNN unit activity during the encoding phase (i.e. Encoding Space) and evaluated its generalization performance in consecutive steps (i.e. Memory Spaces). We reasoned that if the object information is encoded in a subspace that is stable across time (as in a memory slot), the decoders' generalization performance should be high and comparable to its performance during the encoding phase. Contrary to H1's prediction, we found that the decoders do not generalize well (Figure 4b), suggesting that the object information is not stably encoded in a temporally-fixed RNN latent space.

However, we observed that in STMF and MTMF models, the cross-time decoding accuracy is consistently higher during recall (Figure 4b and c). Interestingly, this suggests that the object representation is partially realigned with its original encoding representation when it is retrieved (Fig. 4b).

Next, we examined whether the object encoding space is shared between incoming stimuli, irrespective of the specific object or time (**H2** vs. **H3**; i.e. $E_{(S=i,T=i)} \stackrel{?}{=} E_{(S=j,T=j)}$, for $i \neq j$). We thus fitted classifiers to decode each object property using the hidden activity from the encoding space of each stimulus within the sequence (i.e. decoding $S = i$ from $E_{(S=i,T=i)}$), testing it on the stimuli appearing at other time steps (i.e. decoding $S = j$ from $E_{(S=j,T=j)}$). We performed this analysis for all object properties and for all models and all tasks. The validation and generalization accuracies were almost identical (Figure 4d), suggesting a stable encoding representation ($E_{(S=i,T=i)} = E_{(S=j,T=j)}$) consistent with **H2**. In other words, each object in the sequence was encoded according to its chronological age (i.e., when it was placed into memory), regardless of the object property.

Having examined how the RNN latent space allows concurrent encoding, retention, and retrieval of information, we next investigated what transformations underlie the conversion of information from one subspace to another. Specifically, we inquired whether the transformation of feature subspaces across timesteps is stable with respect to the same encoded stimulus (i.e. $\mathcal{T}_i = \mathcal{T}_{i+1}$; see Fig. 4e). As detailed in Appendix .1, we adopted the orthogonal Procrustes analysis to obtain rotation matrix $R_{S,T}$ to characterize the transformation. The orthogonal Procrustes analysis is a statistical shape analysis which discovers simple rigid transformations that superimpose a set of vectors/points onto another. We used this analysis to inquire whether each set of object feature decoders can be rotated to align with the set of decoders for the same object properties, but across time. Thus, the above test can be reformulated as

$$R_{(S=i,T=j)} \stackrel{?}{=} R_{(S=i,T=j+1)} \tag{2}$$

Additionally, we also tested if the transformation is consistent across stimuli within the sequence:

$$R_{(S=i,T=j)} \stackrel{?}{=} R_{(S=i+k,T=j+k)} \tag{3}$$

We first checked whether the feature representation subspace transformation across timesteps are structured (Figure 4f-left). In other words, we evaluated how well the Procrustes analysis can capture the rotation transformation of feature representation subspaces. The reconstructed decision hyperplanes resulting from rotating the original decoders by the Procrustes transformation matrix $R$ were highly accurate (Fig. 4f-right), indicating that a rotation was able to capture the transformation performed by the RNN across time (also see Figure A4).

Next, we tested Eqs. 2 and 3 by swapping the rotation matrix at $R_{(S=i,T=j)}$ with $R_{(S=i,T=j+1)}$ or $R_{(S=i+k,T=j+k)}$ respectively, and plotted the accuracy of reconstructed decision hyperplane for MTMF models in Figure 4g. We reasoned that if these rotation operations were shared across time steps and stimuli, swapping them would not significantly affect the decoding accuracy. Across all model architectures and tasks, we found that replacing $R_{(S=i+k,T=j+k)}$ consistently yields good accuracy, whereas replacing $R_{(S=i,T=j+1)}$ does not. Similar results were obtained in models trained on larger N (N=1-4), stimuli with naturalistic texture backgrounds, and more number of categories and identities (Fig. A6). These results suggest that while the transformation remains consistent across different stimuli, it is not stable over time.

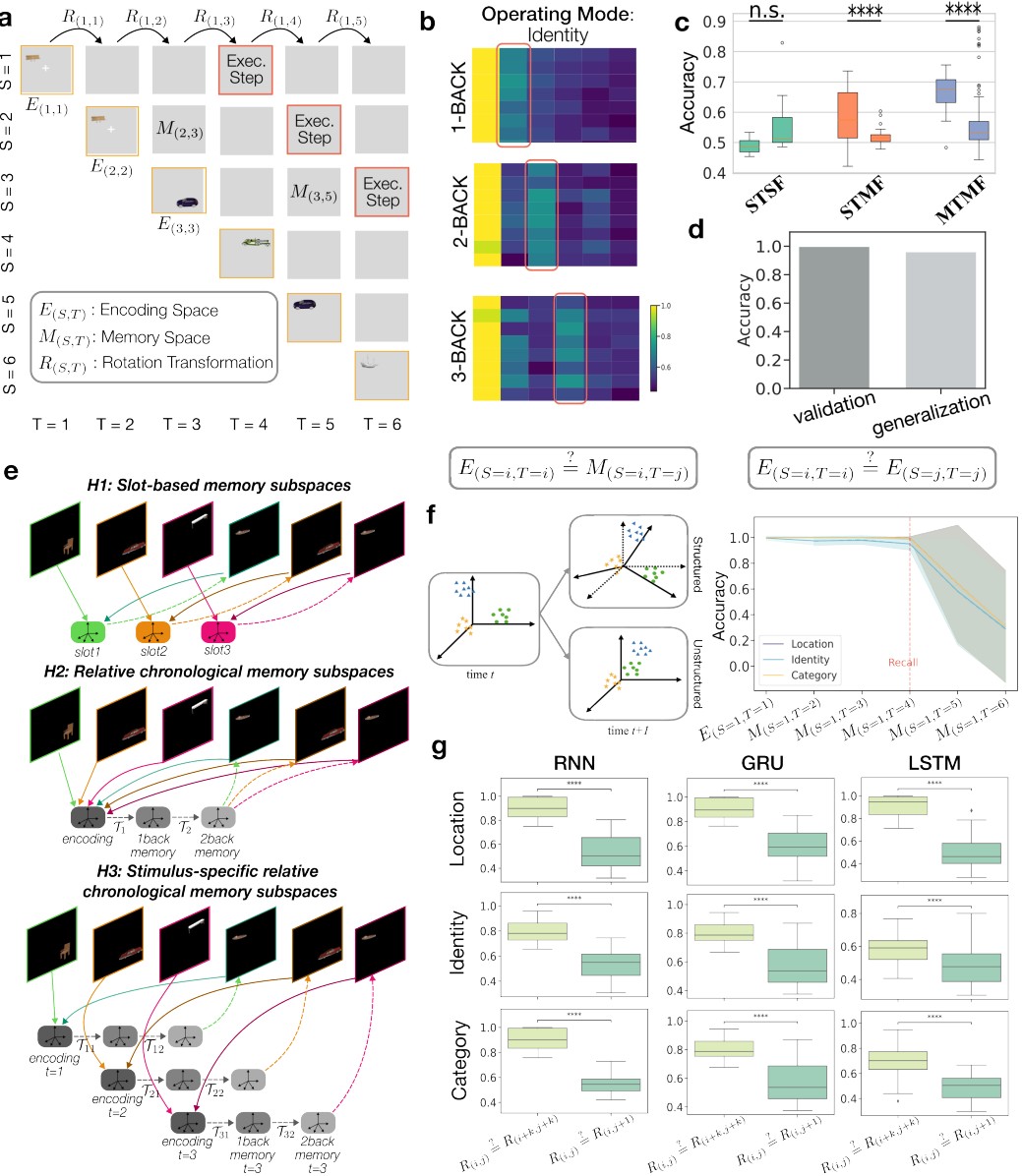

Figure 4: **RNN dynamics during n-back task** a) schematic of the 3-back task for a trial of 6 inputs. Model encodes each observed object in its respective Encoding Space denoted as $E_{(i,j)}$ (diagonal frames with yellow borders). For each stimulus, various object properties are retained over time in their respective Memory Space denoted as $MS$. On executive steps (frames with red borders) model produces a response according to the memory of the stimulus and the newly observed stimulus at that time. b) Decoding accuracy for predicting object identity at different time steps where the decoder is fit to data from the encoding step of a MTMF GRU during 1/2/3-back identity tasks. Red box indicates the executive steps. c) For each model type, we measured the generalization accuracy on executive (left boxplot) and non-executive (right boxplot) steps. d) Decoding accuracy for decoders trained and tested on the same $E_{i,i}$ space (validation) or tested on other $E_{j,j}, j \neq i$ spaces. e) Schematic of the three hypotheses. f) Left: Schematic of the two latent space transformations. Structured transformation preserves the topology (i.e. the transformation can be captured solely by a common scaling factor and a rotation matrix). Unstructured transformation: does not preserve the topology. Right: Decoding accuracy for fitted decoders (solid line) and reconstructed decoders (dotted line) using the rotation matrix $R_{(i,i)}$ from the Procrustes analysis. The small accuracy gap between fitted and reconstructed decoders suggests a structured transformation. g) Decoding accuracy of the reconstructed decoder when the original rotation matrix is substituted with another (indicated by the x-axis labels). Rows and columns corresponds to object properties and MTMF network architectures respectively.

# 5 Discussion and Conclusions

We investigated how naturalistic objects are represented in recurrent models during a dynamic and difficult WM task. In contrast to most prior work that build computational models of WM using abstract categorical stimuli, which typically study WM during a stable fixation/delay period, we trained a range of sensory-cognitive models to perform N-back tasks using naturalistic stimuli. We found that models trained on multiple features and multiple tasks retained object information regardless of their task-relevance. While prior studies investigating WM tasks using RNNs have identified shared representations and dynamical motifs across related tasks [Yang et al., 2019, Driscoll et al., 2022], we found that representations were largely task-specific in gated RNNs such as GRUs and LSTMs. The representational differences between gated and gateless RNNs provides an opportunity for future work to adjudicate between these recurrent architectures and their cognitive significance (i.e., gated vs. gateless) with empirical neural data.

While increased orthogonalization could in theory enable better task performance through the formation of increased separation of object representations in the recurrent module, we found the opposite. Despite reduced orthogonalization of task-relevant features in the RNN, this did not appear to influence task generalization behavior. One possible interpretation of this is that the reduced orthogonalization in the RNN latent space produces a more efficient (lower dimensional) representation. In practice, only a subset of latent need to contain orthogonalized task-relevant representations representations for successful task performance. Having fewer latent dimensions that orthogonalize task-relevant representations could make it easier to decode/read out the correct response information from this latent subspace. Clarifying this distinction further will be important for future work.

Lastly, we found that RNNs solve the N-back task by leveraging chronological memory subspaces to maintain information about different objects distinct. This finding is consistent with the "resource" model of working memory [Alvarez and Cavanagh, 2004, Franconeri et al., 2013], which proposes that memory resources are flexibly distributed across all items. This is contrast to the "slot-based" model [Luck and Vogel, 1997], which suggests that memory is composed of discrete, independent slots for each item. Furthermore, the observed dynamics align with the conceptual framework in Whittington et al. [2023], where different memory slots can transfer information between each other during working memory. Our experiments reveal that similar slot-like subspaces naturally emerge in task-optimized RNNs, with information transfer between them guided by RNN transformations based on task demands. However, as these subspaces are carved out of a shared neural space defined by the RNN units and dependent on chronology, they are not necessarily non-overlapping (unlike that assumed in Whittington et al. [2023]). Such possible overlaps may account for previous findings on the influence of memory load on working memory performance [Ma et al., 2014]. Altogether, these results provide testable predictions to evaluate the neural basis of WM in humans in future work.

# 6 Limitations

Our study has several limitations:

1. **Task-Specific Findings**: Our results are specific to the N-back task structure, and it remains uncertain whether similar computational strategies would emerge in other working memory tasks.

2. **Analysis for Novel Objects**: While we observed reduced performance with novel objects, we did not analyze the representational geometry associated with these stimuli or investigate the reasons for diminished performance. Our use of naturalistic stimuli was intended to avoid imposing the representational geometry typical of abstract inputs, as seen in previous studies (e.g., Piwek et al. [2023], Yang et al. [2019]).

3. **Architectural Constraints**: Our findings are restricted to commonly used RNN architectures, such as vanilla RNNs and LSTMs. Therefore, we cannot make definitive claims about how different neural network architectures might affect representational geometry. Further research could explore whether other architectures yield similar or distinct patterns.

4. **Impact of Network Scaling**: We did not explore how scaling the network size might influence the results. It is possible that increasing the network size could alter the strategies employed by models to perform dynamic WM tasks.

## Acknowledgments

This research was supported by the Healthy-Brains-Healthy-Lives startup supplement grant, the NSERC Discovery grant RGPIN-2021-03035, and CIHR Project Grant PJT-191957. P.B. was supported by FRQ-S Research Scholars Junior 1 grant 310924, and the William Dawson Scholar award. All analyses were executed using resources provided by the Digital Research Alliance of Canada (Compute Canada) and funding from Canada Foundation for Innovation project number 42730.

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

# Appendix

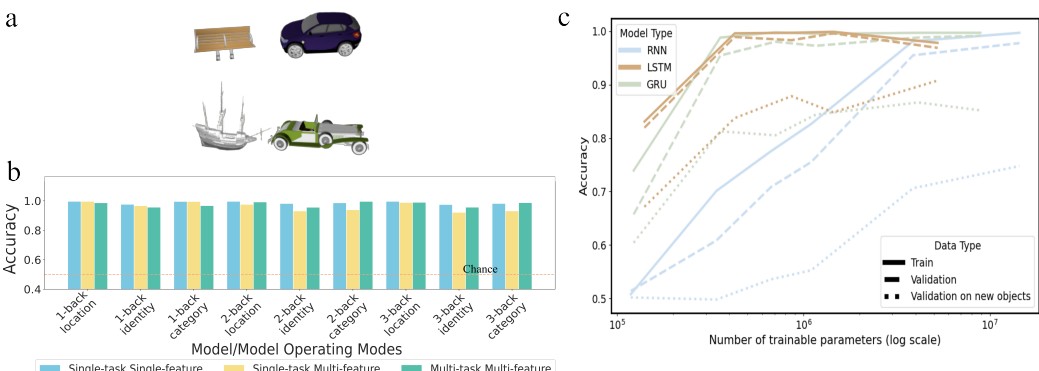

Figure A1: **Stimuli and Model performance** a) rendered stimuli examples from Shapenet. b) 9 task variations of N-back constructed from different choices of task-relevant features ($L, I, C$) and $N$ (1,2,3) index. c) Model performance on train, validation novel angle and validation novel object datasets. Three architectures are tested with various number of hidden size, with the number of trainable parameters indicated on x-axis.

## .1 Methods

**Model training**  We used cross-entropy loss for training, and the identity of the task was encoded in a 6-digit binary format: the first 3 digits represented the one-hot encoding of the feature (e.g., stimulus location, category, or identity), and the second 3 digits represented the one-hot encoding of the n-back choice of n. For the single-task single-feature model, we used the same task identity vector as in the multi-task models. The multi-task multi-feature model typically takes around 4-8k iterations with a batch size of 256, and we cut off training at 14k iterations. The sequence length is fixed at 6 for both the training and validation sets.

**Causal test**  To establish the causal relevance between the decoder-defined subspace and the network's behavioral performance, we perturbed the network's representations by shifting them along the direction of the normal vector to a given decision hyperplane. By passing the resulting hidden states through consecutive timesteps, we computed the probabilities of the three possible actions. We subsampled matched trials and perturbed the hidden states at various magnitudes in the direction of the corresponding decision hyperplane. As shown in Figure A7, the probability of obtaining a match action dropped significantly as the hidden states traversed the hyperplane, while the probability of no action increased. The probability of a non-match action remained largely unaffected, except for an increase in variance as the hidden states crossed the boundary. These results support the causal relationship, indicating that the subspace defined by the decoding analysis is actively utilized by the network in solving the task.

**Decoding analyses**  We consider the latent space of the RNN as a D-dimensional space $\mathbb{R}^D$ where $D$ is the number of units in the RNN model. We used support vector classifiers (SVC) to perform all the decoding analysis. The decoder should be able to classify one feature value from the rest, for example, location bottom left to all the other locations. Each decoder was fitted with activations from either perceptual, encoding, or memory spaces and labels from one of the three possible features $(L, I, C)$ of the stimuli from current or previous time steps[1]. We adopted 10-fold cross-validation as well as grid search to find the best regularization values from $\{0.001, 0.01, 1, 10\}$. All the classifiers reached cross-validated accuracy of $\geq 85\%$.

Each classifier is then described by its normal vector $w$ to its decision hyperplane $d$ and a bias term $b$ .

---

[1]We used Scikit-Learn's implementation of SVC for fitting the classifier parameters.

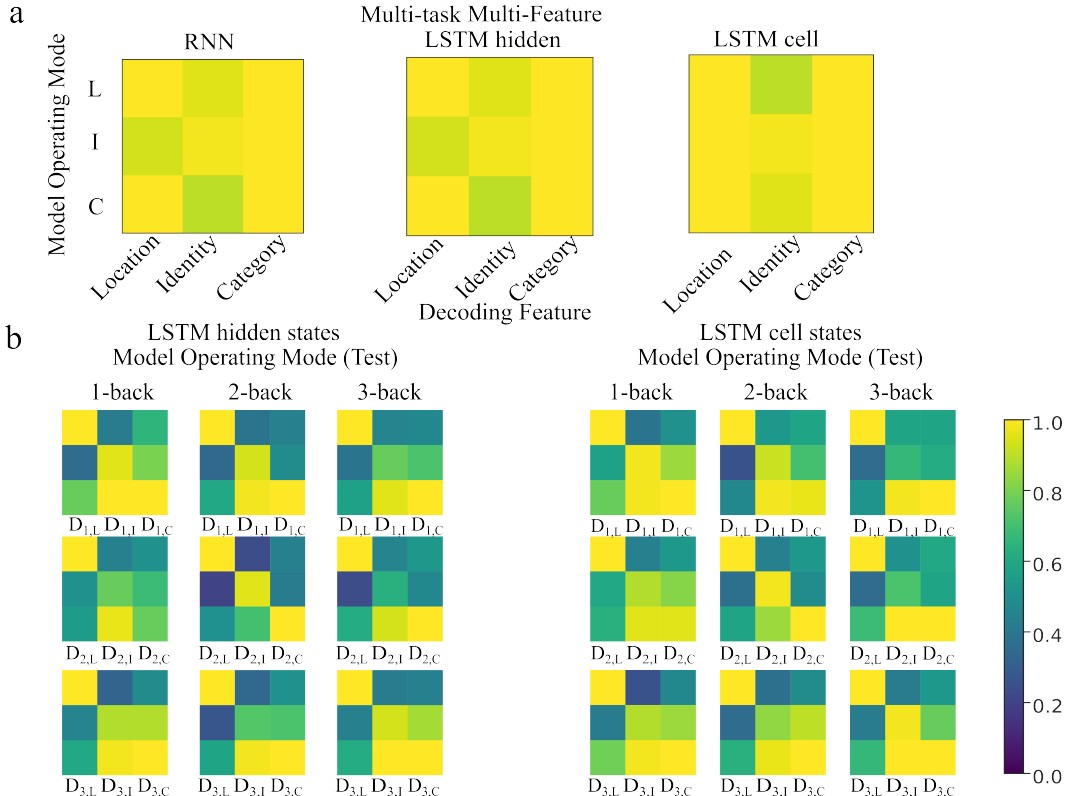

Figure A2: **Object representation efficiency:** a) Similar to Figure 2b, right panel, but for RNN and LSTM trained on MFMT. b) Similar to Figure 2a, but for LSTM hidden and cell states (model trained on MFMT)

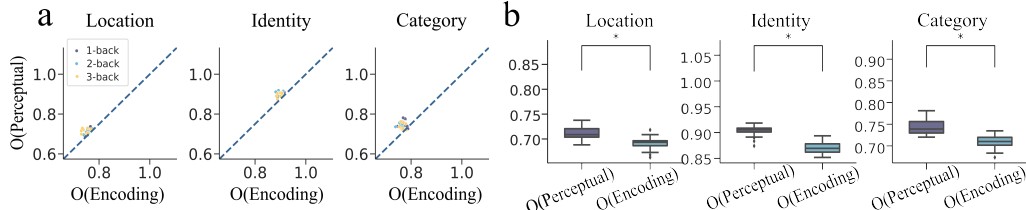

Figure A3: **Orthogonalization:** Similar to Figure 3b, but with PCA to reduce the dimensionality to the same level of the perceptual and encoding space.

$$d = \text{data} \cdot w + b \tag{4}$$

In the Procrustes analyses (see the Procrustes analysis section below), the vector $w$ is derived by rotating the original decoder using the Procrustes transformation and the bias term is obtained from the original decoder. When $d \geq 0$, the corresponding data point is assigned to one class, and to the alternative class when $d \leq 0$.

The set of $N$ vectors $\{\mathbf{w}_1, \mathbf{w}_2, \dots, \mathbf{w}_N\}$ in D-dimensional space $\mathbb{R}^D$, represent $N$ decoders in that space. These vectors span a subspace of $\mathbb{R}^D$ that is the set of all possible linear combinations of these vectors. Mathematically, the subspace $S$ spanned by these vectors is defined as: $S = \text{span}\{\mathbf{w}_1, \mathbf{w}_2, \dots, \mathbf{w}N\} = \left\{ \sum i = 1^N \alpha_i \mathbf{w}_i \mid \alpha_i \in \mathbb{R} \right\}$. .

**Procrustes analysis** Procrustes analysis [Ross, 2004] is a powerful method for identifying shape correspondence that relies on the orthogonality of the rotation matrix. Here, we consider a shape spanned by vertices specified by the normal vectors of decision hyperplanes obtained by classifying

one feature value from the rest. For example, for four possible locations, the shape is defined by four vertices each corresponding to a classifier that discriminates one location from the rest. We calculate such descriptions of shapes at each time step, for each feature, and across all possible tasks for each model. The goal of our analysis is to transform decision hyperplanes obtained under one condition to target decision hyperplanes obtained under another condition. In other words, we want to align source shape with the target shape. To do so, we take the following steps:

- Train the decoders, obtain the normal vectors of the source and target shape, denoted as $w_{source}$ and $w_{target}$

- Standardise $w_{source}$ and $w_{target}$:

$$w'_{source} = \frac{w_{source} - \overline{w_{source}}}{\|w_{source} - \overline{w_{source}}\|_2} \tag{5}$$

$$w'_{target} = \frac{w_{target} - \overline{w_{target}}}{\|w_{target} - \overline{w_{target}}\|_2} \tag{6}$$

We denote $\overline{w_{target}}$ as $\mathbf{b}$ and $\|w_{target} - \overline{w_{target}}\|_2$ as $\mathbf{S}$.

- Perform Orthogonal Procrustes Analysis [Gower and Dijksterhuis, 2004] to align $w'_{source}$ with $w'_{target}$, which returns a rotation matrix ($\mathbf{R}_{source \to target}$) and a global scaling factor ($s$)

- Transform $w'_{source}$ to $w'_{target}$ by

$$w'_{reconstructed} = (w'_{source} \cdot \mathbf{R}_{source \to target})s \tag{7}$$

- Apply the inverted standardization to get the final reconstructed weights:

$$w_{reconstructed} = w'_{reconstructed} \cdot \mathbf{S} + \mathbf{B} \tag{8}$$

The obtained $w_{reconstructed}$ can be used to reconstruct SVC, and the decoding accuracy of the reconstructed SVC can be used as a measure of the alignment quality. In future analysis, we swap $\mathbf{R}_{source \to target}$ by rotation matrix obtained by aligning other pairs of shapes, evaluate the resulting reconstructed SVC's performance and use it to quantify the similarity between two rotation matrices.

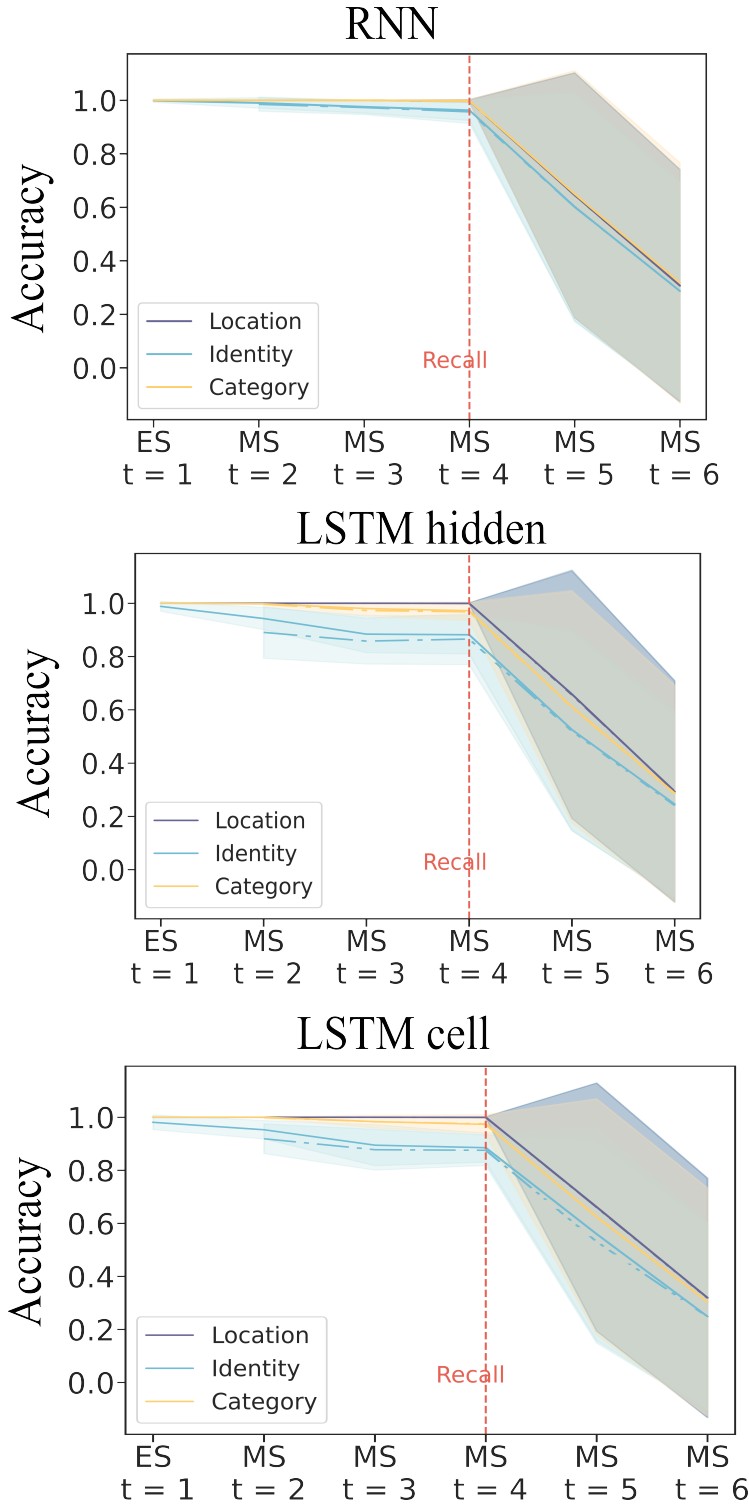

Figure A4: **Validation of Procrustes Analysis:** Similar to Figure 4 g, but for RNN and LSTM trained on MFMT

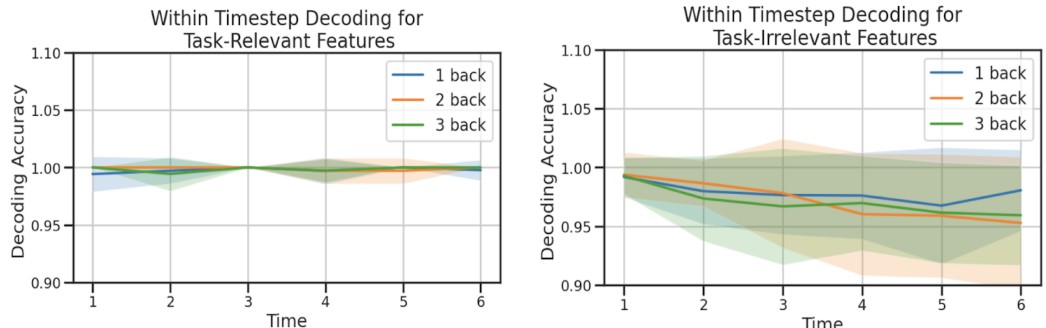

Figure A5: **Within-timestep Decoding Analysis:** At each timestep, we trained SVMs on activations from the recurrent module for task-relevant features (left) and task-irrelevant features (right), plotting the validation accuracies averaged across different feature values. The results shown are for an example GRU model trained on a multi-task, multi-feature task set. As expected, both task-relevant and task-irrelevant features were well represented at their corresponding encoding times. In addition, task-relevant features were more robustly encoded and distinctly separated compared to task-irrelevant ones.

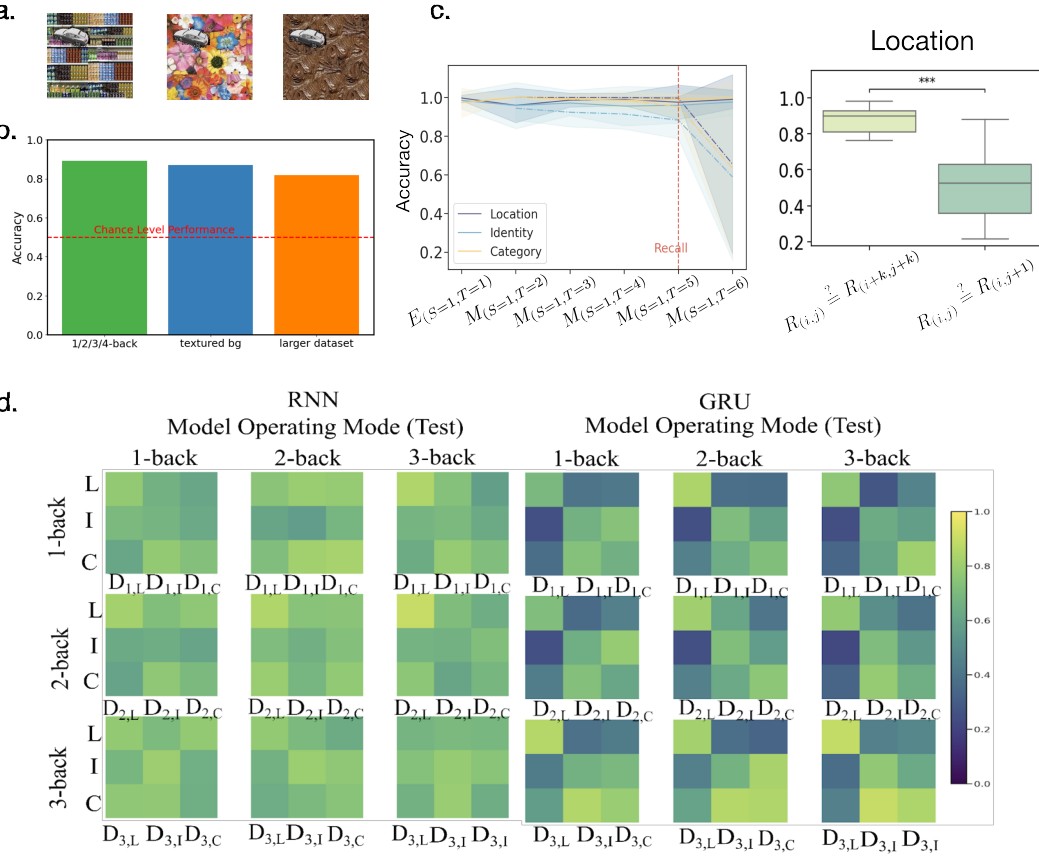

Figure A6: **a) Frames with different natural backgrounds:** We overlaid our original 3D object stimuli on synthesized natural background textures [Efros and Freeman, 2023](`https://github.com/Devashi-Choudhary/Texture-Synthesis`). Three examples of the resulting frames are shown. **b) Task performance for models trained on different datasets:** Accuracies were calculated on the validation set with novel object angles. Multi-task, multi-feature models were trained on more N-back tasks (1/2/3/4-back), visual frames with naturalistic texture backgrounds (texture bg), or tasks generated with a larger stimulus dataset (8 categories with 4 identities each; larger dataset). **c) Rotation transformation analysis:** We identified consistent patterns in the model's response to rotation transformations across various tasks, similar to the results in Figures 4f-g. Left side plot shows an example for the MTMF model performing 4-back tasks. **d) Cross-task decoding analysis:** Consistent decoding results, with higher generalization accuracy for vanilla RNN models compared to gated models.

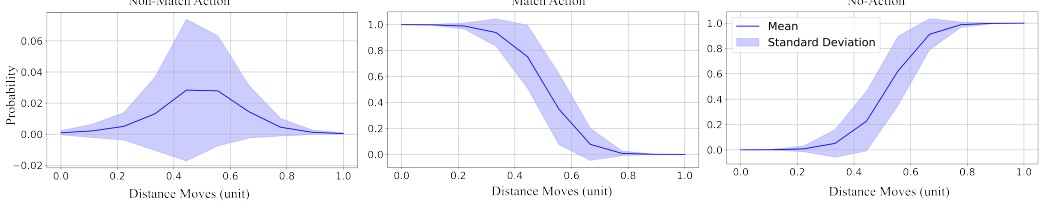

Figure A7: **Causality test:** We subsampled trials with matched actions and trained feature-based two-way decoders. We then perturbed the hidden states, obtained during the presentation of the first stimulus, along the direction of the corresponding decision hyperplane. The distance moved is proportional to the normalized norm vector (x-axis, distance moved in units). The perturbed hidden states were passed through the recurrent module along with the paired stimulus to compute the probabilities of the three possible actions.

