# OpenReview forum: "Geometry of naturalistic object representations in recurrent neural network models of working memory"
_NeurIPS.cc/2024/Conference — NeurIPS 2024 poster_

### Official Review · Reviewer_CRdT · 2024-07-08

**Soundness:** 2
**Presentation:** 2
**Contribution:** 2
**Rating:** 6
**Confidence:** 3

**Summary:**

The paper presents a study of Working Memory (WM) in Recurrent Neural Network (RNN) models.
The main contribution is the study of the latent space dynamics of RNNs during WM-related tasks with respect to naturalistic stimuli, instead of abstract categorical stimuli that are commonly used. The paper analyses gated and non-gated RNNs, and characterises how the geometric properties of their latent spaces change during the trials, as new stimuli are integrated in the WM.

**Strengths:**

* The paper focuses on a more realistic setup than what usually done. It incorporates a “perceptual backbone” given by a CNN, and a “WM backbone” given by a RNN.
* The paper clearly states the tasks (particularly via the plots) and the architectural choices
* The results, being obtained in a more realistic setup that previous works, serve as a basis to formulate testable predictions on biological entities

**Weaknesses:**

* Very small dataset (4 categories, 2 identities per category, 4 possible locations). The N back window also seems limited (how does WM scale for N>3?)
* Stimuli are more realistic, but there is space for improvement (i.e., realistic backgrounds instead of pitch black?)
* The discussion of the three hypotheses in subsection 4.3 is unclear to this reviewer.

If these weaknesses are addressed, the rating may be increased.

**Questions:**

* Does the training from ImageNet transfer to this dataset?
* The normal vectors to the decision boundaries can be taken to be unit norm. In a high dimensional space, however, unit-normed vectors are, with high probability, orthogonal to each other. How does this impact the validity of the orthogonality index?
* It is unclear to this reviewer the reasoning behind the usage of Procrustes analysis for studying the evolution of the decision boundaries.

**Limitations:**

The dataset and the task extent seem to be undersized with respect to what seems reasonably possible. Including larger N-back windows and more objects could be beneficial.

---

> ### Author Rebuttal · Authors · 2024-08-07
>
> We appreciate the reviewer's positive assessment and their suggestions. We address their specific questions and weaknesses below:
> * __Small dataset and limited N-back window__
> We appreciate the reviewer highlighting the limited size of our stimuli and task sets. We would like to clarify that our stimuli dataset includes 4 categories, each with 2 identities. For each identity, there are approximately 20 view angles in the training set and 4 view angles in the validation set. Additionally, we include 2 additional identities for each category in the validation novel object dataset, resulting in a total of 192 stimuli. Despite this dataset size, the model is capable of generalizing to identities with novel angles and novel identities, suggesting that the model is indeed learning (and generalizing) the task. We argue that with larger datasets, the model should retain the same representation geometry as discovered with the given dataset.
> To directly address the reviewer's concerns, we ran several additional experiments. In particular, we trained models on a relatively larger dataset containing 8 categories and 4 identities per category. Due to rendering quality restrictions, we obtained models with comparable training and validation accuracy (~80%), suggesting that the models still captured the task dynamics. We performed a cross-condition decoding analysis (similar to Fig. 2a) and found consistent results, indicating shared representation for vanilla RNNs but not gated RNNs. The relatively low decoding accuracy is a direct consequence of the lower saturation accuracy of the model that will likely be ameliorated with additional training time. Additionally, we further trained models to perform 1-4 back tasks across all 3 features (adding the additional 4-back task setting). Models achieved comparable performance to our original 9-task model, and we were able to replicate the findings from Fig. 4f and g (see attached 1-page PDF Fig 2a). Together, these new results suggested that our findings generalize to broader settings including more diverse object categories and broader task settings.
> * __Suggestion to add realistic background__
> We thank the reviewer for their suggestion and acknowledge that more natural experimental settings are desired and may be important for proper analysis of the neural computations underlying working memory. To address the reviewer’s concern, we ran an additional experiment using stimuli overlaid on synthetically generated natural textures. For this, we synthesized textures as stimulus background using the method from [Texture Synthesis](https://github.com/Devashi-Choudhary/Texture-Synthesis) and overlaid the 3D objects on them to create each frame (see attached 1-page PDF Fig 1c). We then trained models to perform n-back tasks with these new stimuli. We found that models achieved equally well performance levels with comparable speed (see attached 1-page PDF  Fig 1d). In other words, we found that the choice of background does not affect our main results. We plan to include a more detailed analysis of this model in the revised submission (with the full set of analyses).
> * __Interpretation of subsection 4.3__
> Please refer to the general reply for an updated interpretation.
> * __Does the training transfer from Imagenet to Shapenet?__
> As mentioned on line 122-123, our vision backbone consisted of an Imagenet-trained ResNet50 model with fixed parameters. In all our experiments, we used the unit activations of this model as the input to the RNN models. All object features including category, identity, and location were highly decodable from these activations (category: 100.00%, identity: 99.57%, location: 100.00%; 2-fold cross-validation), confirming that the Imagenet trained ResNet50 model generalized to our stimulus set from ShapeNet.
> * __Orthogonalization index interpretation__
> We agree with the reviewer that the high dimensionality of the latent space, specially the perceptual space, could make the comparison potentially invalid. To address the reviewer’s concern, we repeated this analysis by first applying PCA on the activations within perceptual and RNN spaces and equalized the number of dimensions in both. We then calculated the orthogonality measure on the dimensionality matched spaces (perceptual and RNN encoding). This analysis replicated the same findings from the original analysis done without dimensionality matching (see Fig. 1e in the attached 1-page PDF). Lastly, we want to emphasize that the point made in our orthogonality analysis experiments is the difference in orthogonality between the two spaces and we do not directly examine the absolute value of orthogonality, which as the reviewer suggested may be affected by the dimensionality of the latent space.
> * __Rationale behind using the Procrustes analysis__
> We were interested in investigating whether the geometry of object representations is unchanged across different time steps (e.g. between encoding representation and 1st, 2nd, or 3rd memory representations). For this we needed to analyze the likeness of the representational geometry across time steps and for encoding and memory representations. The orthogonal Procrustes analysis is a statistical shape analysis which enables discovering simple rotation transformations that superimpose a set of vectors/points onto another set. We used this analysis to inquire whether each set of object feature decoders can be rotated to align with the set of same decoders at a different time or for a different stimulus.

---

> > ### Comment · Reviewer_CRdT · 2024-08-08
> >
> > I thank the authors for thoroughly answering to my questions.
> > In particular, the discussion of the datasets (their size of the stimuli, the N-back extent and the independence from the background) addresses my doubts entirely, and is convincing.
> > I also appreciate the revised explanations for section 4.3, which were a major point of confusion.
> > Finally, the discussion of the orthogonalisation index was very useful to help my understanding. As for the apparent contradiction in the results obtained when considering the revised definition, the authors’ hypothesis about PCA being unable to capture the high dimensional nature of the perceptual space seems reasonable.
> > As a result of the thorough answers that were provided, I will increase my rating.

---

> > > ### Author Response · Authors · 2024-08-11
> > >
> > > Thank you for your feedback! We appreciate the time you took to review our manuscript and thank you once again for your support  in our work!

---

### Official Review · Reviewer_8PnP · 2024-07-10

**Soundness:** 3
**Presentation:** 2
**Contribution:** 4
**Rating:** 7
**Confidence:** 4

**Summary:**

The manuscript uses various RNN architectures as "model animals" to study the representation and processing of naturalistic stimulus during several working memory (K-back) tasks. Unlike prior work, the current study considers various contexts for the cues, making the representation by the RNNs inherently higher dimensional. They find that when RNNs are required to perform a task with multiple contexts, all RNNs kept track of irrelevant stimuli; yet only vanilla (not gated) RNNs used shared representations across tasks. Moreover, the authors find distinct features are stored in orthogonal subspaces, a finding consistent with the prior experimental literature. They conclude with a study of temporal stability in the face of incoming distractors.

**Strengths:**

- The memory task is complex and requires the RNNs to learn the context.

- The use of naturalistic inputs is in direct contrast and a welcome addition to traditional studies where categorical inputs are used. Though please see [1].

- The observation "These observations challenge the generality of prior studies using vanilla RNNs and categorical inputs in which shared representations and dynamical motifs across related tasks were found [35, 9]." is rather intriguing and very timely, given that [9] just got published as a high-impact publication.

**Weaknesses:**

- Some relevant citations are missing, please see below.
- The presentation, for me, was a bit dense. I was not fully able to follow Section 4.3, though I believe I understood the general takeaways. The authors should work significantly on the presentation.

**Questions:**

I believe the manuscript is novel, convincing, and impactful. Some methodological details are missing, but I am sure the authors can add them during the revisions. Please find my comments/questions below:

- Could you please define what is meant by a latent space, preferably in mathematical terms? I think the authors refer to subspaces defined by the decoder weights, but this should be stated more clearly since latent subspace has different meanings in neuroscience.

- I believe there are at least two seminal works that need to be cited. Please see [1] and [2]; and several citations therein. Specifically, the claim "This is in contrast to prior studies, which primarily studied WM tasks such as delayed-match-to-sample tasks, which don’t evaluate WM maintenance in the face of incoming (and distracting) information." is not correct as both of these works considered attractors. There are more works that should be cited. I believe the authors can find them through a short search, anchoring on these two papers.

- "For this, we trained decoders to predict the value of each object property using the RNN unit activity during the Encoding Space and evaluated its generalization performance in consecutive Memory Spaces." I am wondering if this is a fair test, since the representation may settle into a steady-state after a few time steps. I am not exactly sure how to test this in a way that can account for transient response to die out, so I will leave it to the authors if you wish to address this or not.

Overall, I ended with more exciting questions after reading the manuscript than I started with, which is a mark of a paper that I believe deserves a publication in NeurIPS. I would kindly ask the authors to perform *substantial* edits to improve the presentation so that it is easier to follow the details of Section 4.3.

Citations:

[1] Masse, N. Y., Yang, G. R., Song, H. F., Wang, X. J., & Freedman, D. J. (2019). Circuit mechanisms for the maintenance and manipulation of information in working memory. Nature neuroscience, 22(7), 1159-1167.

[2] Finkelstein, A., Fontolan, L., Economo, M. N., Li, N., Romani, S., & Svoboda, K. (2021). Attractor dynamics gate cortical information flow during decision-making. Nature neuroscience, 24(6), 843-850.

---

> ### Author Rebuttal · Authors · 2024-08-07
>
> We thank the reviewer for their thoughtful response. We also appreciate the acknowledgment of the potentially conflicting results our study highlights in relation to prior works. Below, we respond to individual questions the reviewer raised:
> * __Presentation styles__
> We significantly revised the text in section 4.3 to improve its clarity as the reviewer suggested. Since updating the submission is not allowed during this period, we provide a brief summary of changes: 1) expanding the definition of hypotheses 1-3, adding references to prior related work to each hypothesis; 2) functional interpretation of each hypothesis (Hypothesis 1: sustained WM; Hypothesis 2: dynamic updating; Hypothesis 3: dynamic updating of stimulus-specific memory spaces); 2) adding more references to relevant figures containing definitions; 3) further methodological clarification (time steps and labels used for fitting and testing each decoder); 4) clarified the goal of several analyses (e.g., why did we use the Procrustes analysis, and how was it implemented); 5) added geometric interpretation for each result. We believe that these changes have substantially improved the readability and clarity of the results presented in section 4.3 and hope that will address the reviewer’s concern.
> * __Mathematical definition of latent space__
> The latent space of the RNN refers to a D-dimensional space $\mathbb{R}^D$ where $D$ is the number of units in the RNN model. Consider a set of $N$ vectors $\{\mathbf{v}_1, \mathbf{v}_2, \ldots, \mathbf{v}_N\}$ in D-dimensional space $\mathbb{R}^D$, that represent $N$ decoders in that space. These vectors span a subspace of $\mathbb{R}^D$ that is the set of all possible linear combinations of these vectors. Mathematically, the subspace $S$ spanned by these vectors ( $\text{span} \{ \mathbf{v}_1, \mathbf{v}_2, \ldots, \mathbf{v}_N \}$) is defined as:  $$S = \left\{ \sum_{i=1}^N \alpha_i \mathbf{v}_i \mid \alpha_i \in \mathbb{R} \right\}$$. We will revise the text to make this information and the relation between the decoder weights and the RNN subspace more explicit.
> * __Relevant citations__
> We thank the reviewer for pointing us to these papers. We agree that these two references are highly relevant and we will add them to the updated manuscript and adjust the text accordingly. We also identify the following references relevant to our studies.
>
> * __Questions regarding cross-time generalization analysis__
> We like to first clarify that in our experiments, each model executes exactly one step of computation per input observation and in that sense is different from many prior RNN models used in the literature that involve hundreds of steps within each trial to investigate attractor dynamics in high temporal resolution. For that reason, our experiments do not face the issue of transient responses between two consecutive inputs. In our experiments, we tested the generalization of the decoders up to five steps (the duration of the trial) after observing a stimulus. Thus, if any transient response fades during the trial, our analysis would be able to capture that phenomenon.
>
>
>
>
>
> ## References
> 1. Kozachkov, Leo, John Tauber, Mikael Lundqvist, Scott L. Brincat, Jean-Jacques Slotine, and Earl K. Miller. “Robust and Brain-like Working Memory through Short-Term Synaptic Plasticity.” PLOS Computational Biology 18, no. 12 (December 27, 2022): e1010776.
> 2. Curtis, Clayton E., and Thomas C. Sprague. “Persistent Activity During Working Memory From Front to Back.” Frontiers in Neural Circuits 15 (2021).
> 3. Mejías, Jorge F, and Xiao-Jing Wang. “Mechanisms of Distributed Working Memory in a Large-Scale Network of Macaque Neocortex.” Edited by Tatiana Pasternak and Tirin Moore. eLife 11 (February 24, 2022): e72136.
> 4. Murray, John D., Alberto Bernacchia, Nicholas A. Roy, Christos Constantinidis, Ranulfo Romo, and Xiao-Jing Wang. “Stable Population Coding for Working Memory Coexists with Heterogeneous Neural Dynamics in Prefrontal Cortex.” Proceedings of the National Academy of Sciences 114, no. 2 (January 10, 2017): 394–99.

---

> > ### Comment · Reviewer_8PnP · 2024-08-07
> >
> > I thank the authors for the rebuttal. As noted in my response, I already believe this work to be a suitable contribution to NeurIPS. Both of my concerns were related to the writing, which the authors have committed to addressing. Yet, I have no way of checking the final version, so my current score is the highest I am willing to give due to the poor presentation of the initial submission, which is what I am asked to judge by the guidelines. I wish the authors all the best!

---

> ### Author Response · Authors · 2024-08-08
> **updated sec 4.3**
>
> We understand the reviewer's continuing concern in terms of the clarity of the revised text given that updates to the manuscript are not allowed at this stage. To help the reviewer with their judgement, below we include all of the text from the revised section 4.3, hoping that the reviewer can appreciate the improvements made to the clarity of the new revised section. We highlighted the parts of the text with substantial changes.
>
> # Section 4.3
>
> Having examined the encoding of objects in RNN latent space, we next investigated how RNN dynamics enable simultaneous encoding, maintenance, and retrieval of information. **Performing our N-back task suite required the RNN to keep track of prior objects’ properties as well as the incoming stimuli with minimal interference.** We reasoned that the RNN may implement one of three possible mechanisms to perform the n-back working memory task suite (Figure 4e).
>
> - H1: Slot-based memory subspaces **(Luck and Vogel 1997, Whittington et al. 2023)**. Where the RNN latent space is divided into separate subspaces that are indexed by time within the sequence. Each object is encoded into its corresponding subspace (i.e. slot) and is maintained there until retrieved. **By definition, the subspace assigned to each memory slot is distinct and “sustained” in time.**
>
> - H2: Relative chronological memory subspaces. Where the RNN latent space is divided into separate subspaces that each maintains object information according to their age (i.e. how long ago they were observed, **for example memory of previous observation or prior to last observation). Such a mechanism will require a dynamic process for updating the content of each memory space at each time step during the task.**
>
> - H3: Stimulus-specific relative chronological memory subspaces. Which is similar to the relative chronological memory hypothesis but with independent subspaces assigned to each object. **Each observation in the sequence is thus encoded into a distinct subspace and encoding of each stimulus is in turn distinctly transformed into associated memory representations.**
>
> To identify the hypothesis that best matches the computations performed by the RNN, we analyzed how the RNN **latent subspaces that encodes each observed object property (E(S,T ) ) is transformed across time into memory representations (M(S,T ) )(Fig. 1d).**
>
> We first tested whether object information is maintained in a temporally stable subspace **(i.e. sustained working memory representation)** which aligns with H1 prediction (i.e. E(S=i,T =1) =? M(S=i,T =k), k ∈ {2, 3, 4...}; Fig. 4a). For this, we trained decoders to predict the value of each object property using the RNN unit activity during the encoding **phase (i.e. Encoding Space)** and evaluated its generalization performance in consecutive **steps (i.e. Memory Spaces)**. We reasoned that if the object information is encoded in a subspace that is stable across time **(as in a memory slot),** the decoders’ generalization performance should be high and comparable to its performance during the encoding phase. Contrary to H1’s prediction, we found that the decoders do not generalize well (Figure 4b), suggesting that the object information is not **stably** encoded in **a temporally-fixed** RNN latent space. However, we observed that unlike STSF models, in STMF and MTMF models, the decoding accuracy at the step where the object information needs to be recalled for comparison with the most recent object (i.e. the executive step) is consistently higher than other time steps (Figure 4b and c). This suggests that the object representation is partially **realigned** with its original encoding representation at the step where a comparison is to be made **(Fig. 4b; slightly higher decoding accuracy at the executive step).**
>
> Next, we examined whether the object Encoding Space is shared between all incoming stimuli **within the sequence** (H2) or not (H3) (i.e. E(S=i,T =i) =? E(S=j,T =j)) **– the primary difference between H2 and H3 hypotheses.** We thus fitted classifiers to decode each object property using the hidden activity from **encoding phase (i.e. Encoding Space) of each stimulus within the sequence (i.e. decoding S=i from E(S=i,T =i)), testing it on the stimuli appearing at other time steps (i.e. decoding S=j from E(S=j,T =j)).** We performed this analysis for all object properties and for all model operating modes (i.e. each individual task). The validation and generalization accuracies were almost identical (Figure 4d), suggesting a stable encoding representation **(E(S=i,T =i)=E(S=j,T =j))** consistent with H2. **In other words, each object in the sequence was encoded within the same RNN latent subspace regardless of its order in the sequence.**
>
> *Continued in the next comment*

---

> > ### Author Response · Authors · 2024-08-08
> > **updated section 4.3 cont'd**
> >
> > *Continued from previous comment*
> >
> > Having examined how the RNN latent space allows concurrent encoding, retention, and retrieval of information, we next investigated what transformations underlied the conversion of information from one subspace to another. Specifically, we inquired whether the transformation of feature subspaces across timesteps is stable w.r.t the same encoded stimulus (i.e. Ti = Ti+1; **see Fig. 4e)**. As detailed in Appendix B A.2, we adopted the **orthogonal** Procrustes analysis to obtain rotation matrix RS,T to characterize the transformation. **The orthogonal Procrustes analysis is a statistical shape analysis which enables discovering simple rotation transformations that superposition a set of vectors/points onto another. We used this analysis to inquire whether each set of object feature decoders can be rotated to align with the set of decoders for the same object properties but at a different time step.** Thus, the above test can be reformulated as
> >
> > *Equation (2)*
> >
> > Additionally, we also tested if the transformation is consistent across stimuli **within the sequence,** that is to say:
> >
> > *Equation (3)*
> >
> > Before delving into testing 2 and 3, we first checked whether the feature representation subspace transformation across timesteps are structured or not (Figure 4f, left). In other words, we evaluated whether Procrustes analysis can capture the rotation transformation of feature representation sub- spaces **in the first place. The reconstructed** decision hyperplanes **resulting from rotating the original decoders by the Procrustes transformation matrix R were highly accurate (Figure 4f-right) which indicated that a rotation operation was able to well capture the transformation performed by the RNN across time steps** (also see Appendix A.1. Figure A4). Further, we tested 2 and 3 by swapping the rotation matrix at RS=i,T=j by RS=i,T=j+1 or RS=i+k,T =j+k respectively and plotted the accuracy of reconstructed decision hyperplane for MTMF models in Figure 4g. **We reasoned that if these rotation operations were shared across time steps and stimuli, swapping them should not significantly affect the decoding accuracy.** Across all model architecture and model operating modes, we found that replacing RS=i+k,T =j+k **consistently yields** good accuracy, **whereas replacing** RS=i,T =j+1 **does not. These results** suggest that while the transformation remains consistent across **different** stimuli, it is not stable over time.

---

> ### Comment · Reviewer_8PnP · 2024-08-08
>
> Dear Authors,
>
> I understand your desire and ambition to continue the discussion to increase my current score. However, likely unknowingly, you are utilizing a rebuttal strategy known as "wearing down the reviewers." Such a strategy might work for less experienced reviewers, but I simply used it to update my priors with more evidence for a bad behavior I suspected (more on this later). In my latest response, I made it crystal clear that 6 was the highest score I am comfortable giving, the best response to this would be to thank me for my time and wish me well as well.
>
> Since you want to continue, let me explain why I cannot go beyond 6. The initial submission feels incomplete, as if it is the version that you could get done until the deadline (also the assigned number of 20855 suggests it was one of the last submitted papers).
>
> To some extent, I assigned a solid chance for it being a placeholder for the extensive revisions you planned to perform after the deadline has passed. As several reviewers noted, methodology is incomplete, results are rushed, and several definitions missing. It is *inappropriate* to submit a half-finished work to a conference, in which reviewers are already overburdened. It is even more inappropriate to wear those reviewers down with extensive revisions. Thus, though this work was initially appropriate for a poster thanks to the novel science, the possibility of it being a placeholder prevented me from giving a score higher than 6 for a spotlight recommendation.
>
> Now, with your latest response that seems to add major revisions to your work, my priors are updated and I now more strongly suspect that this is indeed what happened. I do not believe this type of behavior should be rewarded, but **I will leave it to the AC to decide if such major revisions would be appropriate for the NeurIPS conference.**
>
> Due to the reasons described above and my increased suspicion, I will **decrease** my score down to 5. To be perfectly clear and candid, my score can only go down from here, not up. I hope you will take this as a learning experience for future and once again wish you all the best.

---

> > ### Author Response · Authors · 2024-08-11
> >
> > Thank you for your candid feedback and for taking the time to review and discuss our work. We respect your decision and will take your feedback into account for our future work.

---

> > > ### Comment · Reviewer_8PnP · 2024-08-12
> > >
> > > Dear Authors,
> > >
> > > It seems to me that my message went across. As initially noted, I did really like your work. After having gone through other reviews in my batch, I had a little bit more time to look at your responses to me and other reviewers. While I agree with Reviewer VPbf that additional controls could strengthen your study, I don’t believe they are essential for publication. Future work will ultimately determine whether your findings generalize under further controls. I feel more confident that you will be able to address the issues I raised in my original review, so I will gave you the score that you requested.

---

> > > > ### Author Response · Authors · 2024-08-13
> > > > **Thank you!**
> > > >
> > > > Thank you for taking the time to review our responses again and for your thoughtful feedback. We're glad our message was communicated clearly and that you found our work valuable. We also appreciate the score adjustment! Thanks once more for your time and effort in reviewing our work.

---

### Official Review · Reviewer_YsNL · 2024-07-10

**Soundness:** 3
**Presentation:** 2
**Contribution:** 3
**Rating:** 7
**Confidence:** 4

**Summary:**

This paper examines the mechanisms of working memory in recurrent neural networks (RNNs) trained with naturalistic objects and N-back tasks, a classic task in cognitive and neuroscience. The aim is to study more complex, ecologically valid stimuli than the abstract categorical input that previous studies of working memory in RNNs have used. Additionally, the choice of N-back tasks provides a setting in which encoding of new memories, maintenance of prior ones in the face of distracting stimuli, and retrieval must all be balanced. For example, in a 2-back task, I must maintain a memory of the stimulus I saw 1 trial ago, and protect it from interference, as I encode the current stimulus and retrieve the stimulus from 2 trials ago to compare them. Here, the authors investigate what strategies RNNs use to dynamically maintain this object information. Moreover, they investigate RNNs' ability to flexibly switch between multiple tasks (1-, 2-, or 3-back tasks) as well as multiple object properties (identity, category and location).
The findings can be summarized as follows:
1. Information about both task-relevant and task-irrelevant object properties (as measured through a linear classifier's decoding accuracy from the latent vector) is present in multi-task RNNs, but not in single-task RNNs, where only the single property that is relevant for the task is decodable.
2. Information about object properties is orthogonalized in the RNNs' latent space, meaning that the normal vectors of the hyperplanes separating different values of the features used in the task (identity, category and location) are close to orthogonal.
3. In multi-task settings, a vanilla RNN encodes object properties in a subspace that is shared across tasks, while gated RNNs (LSTM and GRU) keep separate subspaces for different tasks.
4. In N-back tasks that require to simultaneously encode, maintain, and retrieve information in working memory, RNNs implement a shared encoding space for all stimuli that appear for the first time, and they do not maintain consistent codes for different objects across time.
5. The transformations of feature subspaces across time can be well-approximated by a rotation matrix, and they are consistent across stimuli, but not stable over time. In other words, the rotation matrix that predicts the transformation of one stimulus from one timepoint to the next can predict the transformation of another stimulus at equivalent processing stages (e.g. between the initial encoding of the stimulus and the next timestep) but it cannot generalize to other processing stages.

Overall, this paper is a thorough investigation of the mechanisms underlying working memory in RNNs in a complex, ecologically valid task setting.

**Strengths:**

- The paper studies a class of tasks (N-back tasks) that elegantly combines different requirements of working memory. It thus allows to study several simultaneous operations, providing a considerably more complete picture relative to previous work. It also uses realistic stimuli, providing a closer approximation to complex tasks used to study working memory in humans and animals.
- The "related works" section provides a concise but informative overview of previous work, including the study of working memory in both biological and artificial systems.
- The experiments are creative and sound, and provide a detailed picture of the computational solutions found by RNNs to solve N-back tasks.

**Weaknesses:**

- The paper contains several typos: for example, at line 106 "dynamical dynamics", lines 238-240 "We thus fit classifiers to decode each object property using the hidden activity from Encoding Space of the stimuli first appears at one timestep testing it on stimuli first appears other time steps.", line 244 "we next investigated how what transformations underlied the conversion of information", line 258: "architecutre". I would recommend the authors to go through the manuscript and correct these errors, as they make reading the paper unnecessarily harder.
- Several details of the model and training are left out: for example, figure 1c shows a hidden layer between the RNN's latent space and the output. What is the size of this layer? What is the training loss? I imagine it is cross-entropy, but this information should be made explicit. Also, how was the identity of the task encoded? Was a dummy (constant, or random) task identity vector also provided to the single-task single-feature model? How many iterations was the model trained for? How long were the sequences in the training and validation sets?
- The generalization to novel object views and novel object instances is briefly mentioned, but not developed any further. The authors should add an explanation of what can be inferred from these results, and how they related to other findings in the paper. Otherwise, I see little reason for including them.
- I am not sure what conclusions can be drawn from the orthogonalization found in the RNNs' latent space. The authors seem to attribute it to the dynamical encoding of stimulus information implemented by the RNNs, but the comparison between the perceptual space (penultimate layer of ResNet) and the latent space of the RNN does not seem appropriate, especially since the RNN is trained, while the ResNet (if I understand correctly) is frozen. It seems natural then that any network (even an additional feedforward layer) explicitly trained for a task would represent stimuli along orthogonal dimensions that corresponds to the task-relevant features. It becomes hard then to interpret the finding of orthogonalized representations.
- While the paper convincingly characterize the subspaces used by the RNNs to perform the N-back tasks, it does not directly test their causal relevance. This seems like it would be relatively straightforward: by shifting the network's representation along the direction of the normal vector to a given hyperplane, it should be possible to "push it" towards giving a specific answer. For example, once the hyperplane separating "car" from other categories has been computed, the RNN's hidden state can be shifted along the normal vector's direction to (falsely) make it output "match" if the object seen N timesteps ago was a car (in a category N-back task). A similar analysis would validate the notion that the subspaces found by the decoding analysis are actually used by the networks in solving the task.
- While the paper contains several analyses looking at _cross_-decoding, in which a classifier trained on a timestep in the sequence is evaluated on another timestep, there is no plot showing the _within-timestep_ decoding accuracy. Such an analysis would be helpful for characterizing the dynamics of the information contained in the representation: for example, is the representation of an object simply maintained in memory, iteratively enhanced, or does it degrade in time? Looking at decoding accuracy over time, or also at the orthogonalization index over time, would help to adjudicate between these possibilities. The fact that representation transformations across timesteps are well approximated by rotation matrices suggests that stimulus discriminability is roughly constant (at least until recall, as shown in fig. 4f) but measuring this discriminability directly as decoding accuracy would help to interpret the finding of rotation-like dynamics in functional terms. As decoding accuracy seems to be in general close to 100%, perhaps a continuous measure such as distance from the classification hyperplane would be more sensitive in revealing these dynamics.
- In general, I feel that the paper does not provide a cohesive story to bind the different results together and interpret them. I think the findings are really interesting, but it is not clear what the take-home message is in terms of what they teach us about working memory in humans, animals or artificial systems. For example, what does the relative chronological memory subspace proposed in H2 (and confirmed by the results) mean in functional terms? What is the functional usefulness of having an encoding-specific subspace shared across stimuli, for example? Or what does the finding of rotation-like dynamics tell us in functional terms? One possible interpretation that I take away from this is that _protection against interference_ is a major principle determining RNNs' dynamical codes. For example, having an encoding-specific subspace can help to maintain the current stimulus separate from the ones being held in memory. Is this also your interpretation of the results? The results, as I said, are very interesting by themselves, but providing your own (even if speculative, that is completely fine) interpretation of what principles underlie these results would substantially strengthen the paper. Relatedly, what is the link between the present results and the broader literature on working memory in neuro- and cognitive science? Do these results allow us to distinguish between theories that have been previously proposed, or do they require a new explanatory framework? I appreciated the "Related works" section, which gave a nice overview of research in this field, but I felt the Discussion was not as thorough in placing the current findings in context.

**Questions:**

No questions, aside from those mentioned in the "weaknesses" section.

**Limitations:**

The paper does not explicitly address the limitations of the current approach. In particular, to what extent the current findings generalize to other working memory tasks, and to what extent they reflect working memory operations in biological systems, is not clarified.

---

> ### Author Rebuttal · Authors · 2024-08-07
>
> We thank the reviewer for recognizing the importance of our study. We have carefully considered the reviewer's comments and have provided the following response:
> * __Typos.__
> We sincerely thank the reviewer for carefully listing all of the typos in our original draft. We will fix them and more thoroughly review the revised manuscript to ensure clarity.
>
> * __Missing details about model training.__
> Thank you for highlighting this. We have included additional methodological (e.g., training details) in our general response to all reviewers. In the revised version of our manuscript, we will include those details, in addition to other missing method details.
>
> * __Discussion on the relevance of generalization to novel object/view angle.__
> The primary reason for our evaluation of novel view/angle generalization was to highlight that the mechanisms of working memory learned by the model were not strictly limited to the specific set of visual features used during training. Hence, we used the same stimuli but with novel views/angles. Nevertheless, we agree with the reviewer that it will be potentially interesting to further examine the representational dynamics of RNNs when considering alternative stimuli (e.g. novel objects). (Note, however, that novel objects would prevent us from evaluating on the n-back identity task, since this requires the same stimulus.) Given the limited time at our disposal for providing a response to the reviews we were unable to perform further analyses in this direction. However, we have added this suggestion as an important future direction in our discussion section.
>
> * __Interpretation of the orthogonalization results.__
> Regarding the update of the orthogonalization analysis and interpretation, we kindly ask the reviewer to refer to our general reply for a detailed explanation due to the limited character count for individual rebuttals.
> * __Causal relevance test.__
> We thank the reviewer for their suggestion. We performed a new analysis to address this point. Specifically, we considered trials from the 1-back location task where two objects had matching location property. We then manually adjusted the hidden state of the network in the direction normal to the decoder’s decision boundary. We then calculated the network’s generated output probabilities as a result of this hidden state adjustment. This analysis revealed that when adjusting the hidden state such that the decoder judges the object location to be different from its original value, the probability of match output is reduced while the probability of non-match and no-action outputs increase (Fig 1c in the attached 1-page PDF). This result thus supports the causal role of the particular encoding of object properties on the network’s generated behavioral output.
>
> * __Within time-step decoding analysis.__
> Following the reviewer’s suggestion, we performed the decoding analyses for each time step separately. The results qualitatively mirrored those we had reported in our Procrustes analysis that showed near perfect decodability up the time step where the object information was necessary for performing the task. Please see Fig. 1b in the attached 1-page PDF.  In addition, as suggested by the reviewer, we acknowledge that given the near-perfect decoding accuracies, using a continuous measure like distance from the decoding hyperplane could provide more nuanced insights into the representation dynamics. We will include the results of this analysis in the final revised version of the manuscript.
> * __Interpretation and relationship to previous literatures.__
> Our goal from the present work was to characterize the way multidimensional object properties are represented in the RNN models of WM and to reveal the mechanisms employed by these models for concurrent encoding, retention, and recall of information according to task demands. We will revise the abstract and the main paper to more clearly reflect these aims and to highlight its importance and distinctness from prior work. Furthermore, our findings in section 4.3 provide evidence against one of the prominent models of working memory, the slot-based model of working memory (Luck and Vogel 1997, Whittington et al. 2023). Relatedly, our results further show that within the family of n-back tasks, the memory representation is not “sustained” as is commonly suggested by prior studies. We expect that a sustained or dynamic memory representation will largely depend on the task structure (e.g., n-back task vs. a working memory delay with a static fixation).
>
> Luck, S. J., & Vogel, E. K. (1997). The capacity of visual working memory for features and conjunctions. Nature, 390(6657), 279-281.
>
> Whittington, J. C., Dorrell, W., Behrens, T. E., Ganguli, S., & El-Gaby, M. (2023). On prefrontal working memory and hippocampal episodic memory: Unifying memories stored in weights and activity slots. bioRxiv, 2023-11.
>
> * __Limited discussion.__
> We agree that the discussion on the limitations of our work was largely incomplete. To address the reviewer’s concern, we substantially revised the Discussion and Conclusion sections, and included a list of limitations and future directions. This includes additional experiments,  including the reviewer’s suggestion on studying other working memory tasks. In particular, we are planning to perform an additional analysis using a simple delayed match-to-sample task to compare how working memory representations are maintained in the absence of incoming stimuli (i.e., the N-back task has incoming stimuli at every ‘delay’ period). If indeed the rotation-like dynamics are employed to protect against interference in the N-back task, we hypothesize that there may be the lack of rotation-like dynamics across time in a delayed match-to-sample (in which the delay is just a fixation). Though we could not perform this analysis during this rebuttal period, we are aiming to include this analysis in the final version of this paper.

---

> ### Comment · Reviewer_YsNL · 2024-08-13
> **Response to authors**
>
> I thank the authors for their rebuttal, and apologize for the delay in responding. Generally, their responses to my points were helpful in clarifying my doubts. I only have further comments on the following points:
>
> - **Orthogonalization results.** As I had written in my initial review, I wasn't sure how to interpret the orthogonalization finding in the first place, so I also don't know how to interpret the updated finding of *lower* orthogonalization in the RNN. It is indeed counterintuitive, as I would think that training with a task would lead to more orthogonalization along task-relevant dimensions. Either way, I don't think this finding is central to the paper's message, so I believe you could simply remove it.
>
> - **Causal relevance test.** I do appreciate that the authors have run this analysis. However, given the high level of variability in the network's response across trials, I do not think any conclusion can be made from this analysis about the causal relevance of the network's subspaces. The significance of the trends shown in Fig. 1c of the attachment could be simply checked using a measure of correlation between percentages of shift and probability of each given response. I doubt that any of the correlations shown in Fig. 1c will be significant. One possible reason for the high variability might be that the authors subsampled "match" trials and tried to push them towards giving a "mismatch" response. This means that the network's state was pushed towards the "all" direction of the "one vs. all" classifier, which probably does not correspond to any particular category, hence the high variability. It is possible that trying to push a "mismatch" response towards a "match" response (i.e. pushing the network towards a specific category) would lead to more reliable results. However, my intuition might be wrong and I understand that the authors don't have the time to run this analysis. If the authors wish to include this additional analysis as it is, then, they should describe it as inconclusive rather than try to infer any causal effect from it.
>
> - **Within-timestep decoding.** The finding of constant decoding accuracy for task-relevant features (and slightly degrading accuracy for task-irrelevant ones) is quite straightforward, and I think it would make a nice addition to the paper. In particular, it provides a nice contrast to findings in RNNs that had to categorize objects without any memory requirements, such as [1, 2], in which decoding accuracy was found to be increasing across time. This suggests a potential distinction between dynamic encoding strategies in perception and working memory. One thing that is not fully clear to me: does Fig. 1b in the attachment indicate that the accuracy was still close to 100% also several timesteps after the time of the response? If so, this would be interesting to highlight. To make this figure clearer, I would suggest adding vertical lines corresponding to the response times in the different conditions, so that it is clear when the accuracy is being measured before or after the response.
>
> - **Relation to other models.** I thank the authors for their clarification. While I'm familiar with the general idea of slot-based models, I haven't had the time to go through the more explicit model proposed by Whittington et al. (2023). However, from a quick skim of that paper it seems like they do address temporally-structured tasks as well (beyond just delayed response)? Particularly, in their Figure 5, they describe a model that shows some similarity to what the authors propose here, whereby stimuli are encoded in a given slot and then rotated, such that the initial slot can be occupied by the new stimulus to be encoded. I would appreciate it if the authors could clarify the distinction between their model and the one in Whittington et al. (2023). In general, I agree about the importance of across-tasks comparisons, and I am glad that the authors plan to add a comparison with a delayed match-to-sample task. I think that will be a nice addition to the paper.
>
> To conclude, as the authors have clarified several points but not made major changes to the paper, I plan to keep my score. Again, apologies for not leaving enough time for more discussion with the authors, but I believe any further revisions would have been minor anyway.
>
> [1] Spoerer, C. J., Kietzmann, T. C., Mehrer, J., Charest, I., & Kriegeskorte, N. (2020). Recurrent neural networks can explain flexible trading of speed and accuracy in biological vision. PLoS computational biology, 16(10), e1008215.
>
> [2] Thorat, S., Aldegheri, G., & Kietzmann, T. C. (2021). Category-orthogonal object features guide information processing in recurrent neural networks trained for object categorization. arXiv preprint arXiv:2111.07898.

---

> > ### Author Response · Authors · 2024-08-13
> >
> > We thank the reviewer for the detailed feedback. Below are our responses to some of the questions:
> > 1. **Causal relevance test**. Thank you for pointing out possible roots of the observed high variability in the results of this additional experiment. We will perform the updated version of the experiment using 1 vs. 1 decoders and pending results will include them into the revised manuscript.
> > 2. > does Fig. 1b in the attachment indicate that the accuracy was still close to 100% also several timesteps after the time of the response?
> >
> > We would like to clarify that the analysis presented in Fig. 1b of the attached PDF pertains to decoding features of the present stimuli, rather than features of stimuli from memory. As a result, it is expected that the information is decodable irrespective of the timestep.
> > We realized that the reviewer may have asked about memory decoding where the object properties of past stimuli are decoded from each time step. For results specifically related to memory decoding, please refer to Fig. 4f in the manuscript, where the solid line represents the decoding of task-relevant features from memory. The dotted lines that represent the decoding accuracy after the rotation transformation almost exactly overlaps with the solid lines.
> >
> > 3. **Relation to Whittington et al. 2023.** We thank the reviewer for further inquiring about the differences between our findings and those of the model in Whittington et al.. The model proposed in Whittington et al. 2023 assumes that the RNN hidden space consists of a fixed number of “activity slots” with fixed, slot dimensionality and read out weight (reading out of a single slot). Furthermore, the dynamics of the hidden state are limited to identity or zero matrices that can be used to create an exact copy of the contents from one slot to another. While there are conceptual similarities between this theory and our findings from task-optimized RNNs, our findings highlight that 1) strict assumptions on the dimensionality of the subspaces, readout weights, and the dynamics need not be fixed and suitable values could emerge through learning on tasks. 2) Moreover, the proposed mechanism in Whittington et al. is insufficient for performing tasks that require comparison of multiple stimuli (e.g. n-back task) as in such situations the network needs to produce an output by comparing its memory content with perceptual inputs but the proposed model consistently reads out of a single memory slot. We will extend the discussion about the differences and similarities between that work and ours in the revised version of the manuscript.

---

### Official Review · Reviewer_VPbf · 2024-07-13

**Soundness:** 2
**Presentation:** 1
**Contribution:** 3
**Rating:** 5
**Confidence:** 4

**Summary:**

This work involves training RNNs on the n-back task with naturalistic images fed into the RNN through a CNN front-end. Through a large number of permutations comprising different task requirements, different combinations of tasks and different architectures, the authors study how task relevant and task irrelevant information is processed in the RNN back-end.

**Strengths:**

The analysis is principled and rigorous. A large number of model-task permutations have been trained, which all support the same conclusions.

**Weaknesses:**

This work is critically lacking the control experiment of simply training an RNN without natural images or CNNs. The aim of this work is to understand how RNNs process natural images in working memory, but I feel that these results (such as representational orthogonalization) can be reproduced if you simply use randomly generated vectors as inputs. I foresee that this would be a logical doubt from many readers, including myself.

One reason to think this way is that the output of the final layer of a pretrained CNN is essentially uninterpretable (and hence no different from being a random vector of numbers) beyond how it can be clustered according to its softmax boundaries. This is especially so when CNNs have location invariance due to their shared convolution across space, and produce very similar outputs at the final layer (from which the RNN is connected to) for images from the same category.

**Questions:**

There seems to be a lack of details in the methods. For example, are the models only trained/tested on 6 time steps?

**Limitations:**

In the checklist, the authors claim that "We point out the limitations of our model and analysis methods in the discussion." But I really cannot find any sentence in Section 5 that describes any limitation of this work at all.

---

> ### Author Rebuttal · Authors · 2024-08-07
>
> We thank the reviewer’s recognition of the strengths of our manuscript, particularly noting the principled and rigorous nature of the approach. We are glad that the extensive training across various model-task permutations robustly supported our conclusions. In response to the weaknesses and limitations the reviewer highlighted, we carefully provide detailed responses addressing each point:
> * __Lack of control experiments and whether naturalistic stimuli can be replaced by random vectors.__
> We used a pretrained CNN (on ImageNet), which was trained to categorize objects. This suggests that the output layer of the CNN is, overall, lower-dimensional than its inputs (given that the number of objects classified is fewer than the number of total samples in ImageNet). Nevertheless, to verify this intuition on images sampled from our dataset, we measured the dimensionality of the inputs relative to the dimensionality of the CNN output layer, and the dimensionality of the RNN space. As the Reviewer suggested, we also used randomly sampled vectors (the same number as the number of images in our dataset), and measured the dimensionality of the stimulus vectors in the  input space, CNN output layer, and RNN layer. We then compared the dimensionality of our stimulus set vs. randomly sampled vectors for each of these embeddings. We found that the dimensionality of randomly sampled vectors is significantly higher than the dimensionality of the stimuli used in our experiments, suggesting that our results are unique to our specific chosen stimuli and task, and cannot be replaced by using randomly sampled vectors (see fig 1a in the attached 1-page PDF). **Importantly, to directly address the Reviewer’s comment, what this control analysis shows is that the output of the final layer of the pretrained CNN is different from being a random vector of numbers, since the dimensionality of random vectors is higher-dimensional than input vectors from our stimulus set.** Since dimensionality measures the orthogonality between pairs of vectors (stimuli), our analysis shows that the degree of orthogonalization differs (and cannot be directly reproduced) with random vectors.
> * __Lack of details in the Methods.__
> We thank the reviewer for bringing this point to our attention. All models were trained and tested on trials of the same length (6 time steps). Moreover, we have included additional methodological (e.g., training details) in our general response to all reviewers. In the revised version of our manuscript, we include those details, in addition to other missing method details.
> * __Limitations in the discussion.__
> Thank you for bringing this issue to our attention. We have substantially revised the Discussion and Conclusion sections of the paper, expanding on the current study’s limitations and likely future directions. Since updating the submission is not allowed during this period, we provide a brief summary as follows: 1) our findings are limited to the N-back task structure and it’s unclear whether similar computational strategies will emerge in other tasks requiring working memory. However, we provide specific predictions and further experiments that may lead to the discovery of more general principles used by RNNs for different WM tasks (e.g., in a simple delayed-match-to-sample task with only a fixation for the delay, we hypothesize that there will not be a rotation of representations through time); 2) we haven’t considered novel objects in our analyses. We reported reduced performance for novel objects but did not examine the representational geometry with those stimuli or the possible reasons for the diminished performance. We welcome any further suggestions from the reviewer; 3) Our findings are limited to the commonly-used RNN architectures, and we cannot make definitive claims about the representational geometry resulting from other neural network architectural choices. Additionally, we have not examined how scaling the network affects the results; 4) Our results suggest a possible hypothesis about working memory, but further investigation with real neural data is needed to validate this. It would also be valuable to explore the relationship between architectural choices and experimental outcomes.

---

> > ### Comment · Reviewer_VPbf · 2024-08-10
> > **Early response**
> >
> > I thank the authors for the response, and I will reply again with my final decision after carefully reviewing the results. I just have a simple question for the authors:
> >
> > The authors stated in another reply to another reviewer that the total stimuli count is 192. Can I ask how the dimensionality of random vectors, in Figure 1A of the rebuttal PDF, is more than 4000?

---

> > > ### Author Response · Authors · 2024-08-11
> > >
> > > We initialize the random vector input to match the dimensions of our naturalistic input, 224×224×3, drawn from a standard Gaussian distribution. For the input frames, we place one of the 192 possible stimuli in one of the four location quadrants, introducing slight variations from frame to frame to add randomness. Specifically, we add a Gaussian random variable to the sampled central location coordinates within the selected quadrant when placing the stimulus on the frame. As a result, each frame with the same stimulus in the same quadrant is not identical in pixel space. To address input dimensionality, we perform PCA on both the random vector inputs and naturalistic inputs, selecting the first k principal components that account for at least 90% of the variance. This analysis is conducted using 5,000 task trials, which explains why the dimensionality of the random vector inputs exceeds 4,000.

---

> ### Comment · Reviewer_VPbf · 2024-08-12
> **Controls**
>
> It sounds like there are 192 $\times$ 4 different stimuli instead of 192? I will just arbitrarily assume its 192 in my argument below but replace it with the correct number if I'm wrong.
>
> I would normally consider what the authors did as a strawman argument. Noise aside, 192 stimuli identities would have a maximum of 192 dimensions. Comparing them to 5000 randomly generated vectors makes absolutely no sense at all. But I would like to give the authors the benefit of the doubt and clarify myself:
>
> Let $x$ be the final layer of the CNN. $W_{xh} x$ is the input to the RNN, where $W_{xh}$ is the set of weights that transforms CNN outputs into RNN inputs. I am saying if you replace $x$ with 192 random vectors, each representing one identity of your stimuli, and $W_{xh}$ is trainable, it is possible to get the same results. Nothing before $W_{xh} x$ matters.
>
> The title of the paper contains the phrase "naturalistic object representations in recurrent neural network models", and I am challenging the very fact that the input to the RNN is not naturalistic. In fact, prior literature has shown that WM orthogonalization can occur with random inputs [1]. It is extremely important that the authors distinguish their model from a model that does not have a CNN, yet capable of producing the same phenomenon.
>
> [1] https://doi.org/10.1371/journal.pcbi.1011555

---

> > ### Author Response · Authors · 2024-08-12
> >
> > **Regarding the total number of stimuli**: We rendered 192 stimuli from the 3D dataset ShapeNet. These stimuli were then positioned in one of the four quadrants on the frame to generate N-back task frames, as location is a key feature of interest.
> >
> > **192 stimuli identities and dimensionality**: We believe the reviewer may be referring to categorical encoding (binary encoding/one-hot encoding etc) when suggesting that 192 stimuli identities would have 192 dimensions. However, our definition of dimensionality pertains to the neural representation space. As illustrated in Figure 1a of the attached PDF, the dimensionality of the CNN embeddings is 92, not 192, as the total number of stimuli identities might suggest. This reduction in dimensionality reflects the CNN's invariance to viewing angles, mixed selectivity, and the hierarchical nature of features like category and identity. Manually constructing an abstract code that encompasses all these aspects is challenging. Additionally, we use CNN embeddings rather than hard-coded stimulus identities as input to the RNN because CNNs pretrained on ImageNet exhibit significant representational similarity to the human visual system (Yamins et al., 2014 etc).
> >
> > Furthermore, we would like to emphasize that the models we trained can generalize to novel view angles within the same identity and to novel identities within the same categories. This suggests that the model is truly learning the task rather than overfitting the data. We find it unlikely that this level of generalization would emerge from a model trained with an abstract input space.
> >
> > Moreover, we are interested in the relative orthogonalization level within the perceptual and encoding spaces. As shown in the updated Figure 2e of the attached PDF, RNN representations are orthogonalized, with the orthogonalization index for the encoding space nearing 1. However, a critical point we wish to address is whether RNNs orthogonalize task-relevant feature spaces more due to task requirements compared to CNN space. If we were to use hardcoded stimulus identity encoding as RNN input, such a comparison analysis would not be feasible. Unlike Piwek et al. (2023), who utilized feature-sensitive neurons with circular normal tuning functions to encode categories or identities, our feature spaces are not one-dimensional continuous variables.
> >
> > Our claim differs from Piwek et al. (2023), which demonstrates an orthogonal-to-parallel transformation of the cued versus un-cued color planes from the pre-cue to retro-cue period. In contrast, our feature spaces remain relatively orthogonal in both the perceptual and encoding spaces. The input construction in Piwek et al. (2023) assumes an orthogonal relationship between the concatenated encodings of two locations and the colors (using 17 feature-sensitive units for each possible location, following a circular normal tuning function). Therefore, the orthogonal representation at the beginning of each trial is expected. In our case, we did not assume that CNN embeddings are orthogonal, and any representational geometry we observe is an emergent property.
> >
> > Finally, our use of naturalistic encoding is not intended to suggest that results obtained using abstract encoding are incorrect. Rather, it aims to remove input hypotheses that are manually introduced into model design. While Piwek et al. (2023) could potentially achieve similar representational dynamics using a model that takes RGB encoding of color conjugated with location as input, our naturalistic approach allows us to explore questions beyond the constraints of input encoding design. We agree with the reviewer that some of the same findings may be replicated if non-naturalistic inputs (such as random vectors) were to be used as input to the RNN but we respectfully disagree with the reviewer that the input to the RNN in our experiments is not naturalistic. In our experiments, the RNN receives a vector representation of a naturalistic stimulus on every step which is processed by a biologically-plausible vision neural network. We remain open to further comments by the reviewer.
> >
> > We hope this response addresses the reviewer’s concerns, and we look forward to further feedback! Thanks!
> >
> > [1] Yamins, Daniel LK, et al. "Performance-optimized hierarchical models predict neural responses in higher visual cortex." Proceedings of the national academy of sciences 111.23 (2014): 8619-8624.

---

> > > ### Comment · Reviewer_VPbf · 2024-08-12
> > > **Last comment**
> > >
> > > I thank the authors for the response. Also thanks for the clarification on the number of inputs, although it really should not be this tough to figure this out. As I carefully said in the response, the *maximum* dimensionality is 192, nothing about the actual dimensionality (in the extreme case you can put everything in one line and it will be 2-D and there will still be a decoder that separates them). I summarize the author's points below, for additional discussion with reviewers later.
> > >
> > > 1. The authors cited the Brainscore series of works to justify that the output of CNNs are biologically plausible, hence the input to the RNN is biologically plausible.
> > >
> > > 2. The model can generalize to novel view angles which abstract inputs cannot.
> > >
> > > 3. Experiments such as Figure 2e (rebuttal PDF) cannot be done with abstract inputs.
> > >
> > > 4. There are some mechanistic differences in the methods of this work and Piwek et al. 2023
> > >
> > > 5. The authors concede that some findings can be replicated with abstract inputs but disagree that their input is not naturalistic.
> > >
> > > (Points 1 and 2) I will give a very simple example. Suppose the task has 2 categories, cats and dogs. Here is how the CNN works:
> > >
> > > Case 1
> > > - Natural image of cat is the input
> > > - The final layer of the CNN is [12.534, -1.245]
> > > - After softmax, the output is [1,0] implying the classification is a cat
> > >
> > > Case 2
> > > - Natural image of dog is the input
> > > - The final layer of the CNN is [-3.142, 4.123]
> > > - After softmax, the output is [0,1] implying the classification is a dog
> > >
> > > The authors are arguing that the vectors [12.534, -1.245] and [-3.142, 4.123] represent naturalistic inputs of the dog and cat into the RNN. The model can "generalize" to new angles because the CNN will give the correct classification regardless of view angles. In addition, BrainScore looks at the *convolutional* layers, not the post-flattened layer being biologically realistic.
> > >
> > > (Point 3) The authors wrongly assume the results in 2e cannot be replicated with abstract inputs, because the trained weights $W_{xh}$ can manifest the same results (and its even emergent!). Once again this could have been avoided if the control had simply been done (I understand it's too late now).
> > >
> > > (Point 4) I acknowledge the differences (clearly the simulations cannot be exactly the same when the tasks are different), but the point is that RNNs in literature have been trained on abstract inputs, and this paper specifically highlights this as a problem in lines 41-45. If this is the stance that this work takes, and even puts "naturalistic" in the title, then this work must prove itself to give different results using abstract encoding.
> > >
> > > (Point 5) I can't argue with the author's opinions. I have made my case and the other reviewers should as well after reading the series of exchanges here.
> > >
> > > For reasons above, I will keep my score.

---

> > > > ### Author Response · Authors · 2024-08-13
> > > >
> > > > We sincerely appreciate the reviewer’s continued engagement in discussing our work. We would like to clarify some final points that had been previously raised.
> > > > 1.  We would like to clarify an important aspect of our experiment that may have caused some confusion. Specifically, **we use the output from the layer 4.2 ReLU, which is distinct from the logit/softmax layers.** (The dimensions of the ReLU layer are 512 * H/32 * W/32, where H and W represent the height and width of the image, respectively, with H = W = 224. As in contrast, the dimensionality of the logit layer is 2014*C, where C is the number of categories). We mention this because the examples provided by the reviewer in their recent comments seem to assume a vector representation from the logit/softmax layers.
> > > >
> > > > 2. > If this is the stance that this work takes, and even puts "naturalistic" in the title, then this work must prove itself to give different results using abstract encoding.
> > > >
> > > > We would like to clarify that in the present work, we are not claiming the representational geometry is different from those in RNNs trained with abstract inputs as we are not explicitly comparing the two cases. Our title contains the term “naturalistic” because we simply used naturalistic inputs to investigate how those multidimensional representations are encoded and memorized by RNNs. Regarding the reviewer’s comment about our statement on lines 41-45: *“While training models with such categorical stimuli is easier in practice, the resulting models offer limited insights into how real world, naturalistic stimuli (which are encoded in high-dimensional spaces) are processed by these models.”* The motivation behind lines 41-45 was to simply point out that the understanding of how networks function with naturalistic stimuli in a generalizable manner are incomplete, and we investigate this matter throughout the manuscript. However, in the light of the discussion, we will revise these sentences in accordance with the reviewer’s suggestions.
> > > >
> > > > 3. > The authors wrongly assume the results in 2e cannot be replicated with abstract inputs
> > > >
> > > > We apologize that we initially misunderstood the reviewer’s request for the additional analysis. While we agree that the analysis in Fig. 2e of the rebuttal can be theoretically replicated for RNNs trained on abstract inputs, the measure of orthogonality will be 1 in the input space (to the RNN), which is already at ceiling. The reason is because abstract (i.e., one-hot) vectors are orthogonal to each other, which would make it trivial to compare orthogonality between the abstract input space (with orthogonality at 1) with the RNN’s latent space. This triviality is not the case for naturalistic images, particularly since our task requires generalization to view angles and stimulus identities.
> > > >
> > > > We once again thank the reviewer for their time.

---

> > > > > ### Comment · Reviewer_VPbf · 2024-08-13
> > > > > **Thank you**
> > > > >
> > > > > I thank the authors for the clarification. I was working with the "default" CNN-RNN architecture which involves connecting the flattened layer of the CNN to an RNN. (Such models are commonly known as convolutional RNNs, and there are easily hundreds of papers on this model if you search "convolutional recurrent neural network" on google scholar). Even in neuroscience this architecture has been proposed, such as in [Xie et al. 2023](https://www.biorxiv.org/content/10.1101/2023.03.30.534982v1.abstract).
> > > > >
> > > > > Now that the authors have clarified that they are, in fact, connecting to a convolutional layer, I can accept that there is some biological plausibility in this model. Convolutions were originally inspired by complex cells in the first place, and the layered structure is analogous to the sequential processing along the visual pathway. In fact, I respect the authors for not following the norm in the name of biological plausibility.
> > > > >
> > > > > I went back to the main text again, and I see that the authors described this as the "penultimate layer" of the CNN, which is more generally referring to either the pre-softmax layer or the pooling layer -- this can be made clearer and emphasized, because this practice (rightfully) detracts from the norm as mentioned above.
> > > > >
> > > > > Overall, I am glad that this has been cleared up. With this clarification, the control I requested is still important but not absolutely critical like I previously insisted. I will increase my score by 1 and I wish the authors the best of luck.

---

> > > > > > ### Author Response · Authors · 2024-08-13
> > > > > >
> > > > > > Thank you for your candid feedback and thoughtful comments throughout the review process. In response to the reviewer's suggestion, we will revise the phrasing from "penultimate layer" of the CNN to "convolutional layer" to avoid any potential confusion in the final revisions. We appreciate your time and effort in helping us improve our work!

---

### Author Rebuttal · Authors · 2024-08-07

We sincerely thank the reviewers for their insightful and positive feedback. We are encouraged by the reviewers’ acknowledgement of the rigorousness and thoroughness of our work (*reviewer VPbf and YsNL*), creativity and soundness of our experiments (*reviewer YsNL*), and novelty and impactfulness of our results (*reviewer YsNL and 8PnP*), while identifying its questions and limitations. Here we point out several common questions raised in the reviews. (We respond to all inquiries in individual responses.)
* **Updated results in Fig. 3b.** We would like to first acknowledge an error in producing the plots in Fig 3b. The measure we used in the original manuscript was $O = \| \text{abs}(\cos(w_i, w_j)) - I\|_F$ with $I$ as the identity matrix. This measure however produced an inverted measure of orthogonality as we should have used $O = \|\text{abs}(\cos(w_i, w_j)) - \mathbf{1}\|_F$ and only considered the upper (or lower) triangle (thus the higher the orthogonalization index, the more orthogonalized the representation). When correcting the measure, our results and interpretation of them were reversed (see fig 2e in the attached 1-page PDF). Both space are highly orthogonalized, however, even when equalizing the dimensionality of the perceptual and RNN space with PCA, the results demonstrated significant reduction in orthogonalization of the latent space in the RNN. We revised our description of the method and results related to this section accordingly. Our current hypothesis for this counter-intuitive result is that the high-dimensional perceptual space may not have been fully captured by PCA. Specifically, PCA might not have adequately represented the relative dimensional differences between the two spaces under comparison. We kindly ask for the reviewers' opinions and insights regarding this updated result.
* **Missing details in Methods.** We appreciate the reviewers’ suggestions and for pointing out the lack of details in some places. There is only one feedforward layer with a softmax activation that projects the hidden state to one of the three possible output actions. We understand that the visualization may have caused confusion, and we will update it accordingly in the revised manuscript. We used cross-entropy loss for training, and the identity of the task was encoded in a 6-digit binary format: the first 3 digits represented the one-hot encoding of the feature (e.g., stimulus location, category, or identity), and the second 3 digits represented the one-hot encoding of the n-back choice of n. For the single-task single-feature model, we used the same task identity vector as in the multi-task models. The multi-task multi-feature model typically takes around 4-8k iterations with a batch size of 256, and we cut off training at 14k iterations. The sequence length is fixed at 6 for both the training and validation sets. In the revised version of the paper, we will include all necessary training details in the supplementary materials. In addition, we will make the code and trained models publicly available.
* **Interpretation of subsection 4.3**. To address the reviewer’s comment, we revised the portion of the text that details the three hypotheses. Since updating the manuscript is not allowed during this period, we present the revised text below:
**H1**: Slot-based memory subspaces (Luck and Vogel 1997, Whittington et al. 2023). Where the RNN latent space is divided into separate subspaces that are indexed by time within the sequence. Each object is encoded into its corresponding subspace (i.e. slot) and is maintained there until retrieved. By definition, the subspace assigned to each memory slot is distinct and “sustained” in time.
**H2**: Relative chronological memory subspaces. Where the RNN latent space is divided into separate subspaces that each maintains object information according to their age (i.e. how long ago they were observed, for example memory of observation from one, two or three time steps ago). Such a mechanism will require a dynamic process for updating the content of each memory space at each time step during the task.
**H3**: Stimulus-specific relative chronological memory subspaces. Which is similar to the relative chronological memory hypothesis but with independent subspaces assigned to each object. Each observation in the sequence is encoded into a distinct subspace and encoding of each stimulus is in turn distinctly transformed into associated memory representations.

---

### Decision · Program_Chairs · 2024-09-25

**Decision:**

Accept (poster)

**Comment:**

In this study, the authors developed sensory-cognitive models, comprising of a convolutional neural network (CNN) coupled with a recurrent neural network (RNN), and trained them on nine distinct N-back tasks using naturalistic stimuli. The representational geometry and dynamics were investigated using latent subspace analysis. The reviewers had several concerns about the motivation/task design/writing. In particular, the reviewers asked for additional control experiments to strengthen the claims here. The rebuttal went very well and all reviewers gave positive comments. I am inclined to accept the paper.